# Enrichment drives emergence of functional columns and improves sensory coding in the whisker map in L2/3 of mouse S1

Amy M LeMessurier[†], Keven J Laboy-Juárez, Kathryn McClain, Shilin Chen, Theresa Nguyen, Daniel E Feldman*

Department of Molecular and Cellular Biology, Helen Wills Neuroscience Institute, University of California, Berkeley, Berkeley, United States

**Abstract** Sensory maps in layer (L) 2/3 of rodent cortex lack precise functional column boundaries, and instead exhibit locally heterogeneous (salt-and-pepper) tuning superimposed on smooth global topography. Could this organization be a byproduct of impoverished experience in laboratory housing? We compared whisker map somatotopy in L2/3 and L4 excitatory cells of somatosensory (S1) cortex in normally housed vs. tactile-enriched mice, using GCaMP6s imaging. Normally housed mice had a dispersed, salt-and-pepper whisker map in L2/3, but L4 was more topographically precise. Enrichment (P21 to P46-71) sharpened whisker tuning and decreased, but did not abolish, local tuning heterogeneity. In L2/3, enrichment strengthened and sharpened whisker point representations, and created functional boundaries of tuning similarity and noise correlations at column edges. Thus, enrichment drives emergence of functional columnar topography in S1, and reduces local tuning heterogeneity. These changes predict better touch detection by neural populations within each column.
DOI: https://doi.org/10.7554/eLife.46321.001

**\*For correspondence:**
dfeldman@berkeley.edu

**Present address:** [†]Department of Otolaryngology, Department of Neuroscience and Physiology, Skirball Institute, Neuroscience Institute, New York University School of Medicine, New York, United States

**Competing interests:** The authors declare that no competing interests exist.

## Introduction

Sensory maps are a defining feature of cortical sensory areas, and may enable efficient local network computations (*Chklovskii and Koulakov, 2004*). In rodents, maps for sensory features in L2/3 of primary sensory cortex exhibit strong tuning heterogeneity between nearby neurons, superimposed on a smoothly varying global topography. In these maps, discrete functional column boundaries are lacking and the neural ensemble activated by a single sensory feature is spatially dispersed. It is unclear whether this dispersed topography reflects sensory experience during development. Appropriate sensory experience is often required for development of topography (*Katz and Shatz, 1996*), and L2/3 maps are robustly plastic to sensory experience (*Buonomano and Merzenich, 1998*; *Feldman and Brecht, 2005*; *LeMessurier and Feldman, 2018*). This raises the question of whether dispersed, salt-and-pepper organization at cellular resolution is a natural feature of L2/3, or may reflect incomplete activity-dependent development due to impoverished sensory experience in rodent laboratory housing.

Sensory experience may be particularly relevant for development of whisker map somatotopy in rodent S1. Whiskers are tactile sensors represented in S1 by a topographic array of cortical columns, each centered on a L4 barrel. In the canonical whisker map, inferred from single-unit recordings, nearly all neurons are tuned for their column's anatomically-corresponding whisker. However, recent studies using cellular-resolution population calcium imaging in L2/3 reveal that while average tuning within each column is topographically correct, tuning of individual neurons within each column is

heterogeneous, with intermixed neurons tuned for either the columnar whisker or local surround whiskers. The set of neurons tuned for one whisker is dispersed across multiple columns in L2/3 without discrete functional column borders (*Sato et al., 2007*; *Clancy et al., 2015*). This globally smooth but locally scattered representation resembles salt-and-pepper maps of orientation tuning and retinotopy in rodent V1 (*Ohki et al., 2005*; *Bonin et al., 2011*) and frequency tuning in rodent A1 (*Kanold et al., 2014*).

Rodent housing conditions were standard or not reported in prior studies of dispersed, salt-and-pepper maps, but may affect map structure (*Andermann and Moore, 2006*). In particular, environmental enrichment and exercise increase the amount, complexity and salience of sensory stimuli, and promote cortical development (*Diamond et al., 1976*). Enrichment sharpens single-unit receptive fields and refines average map topography in forepaw S1 and A1 of young rodents (*Coq and Xerri, 1998*; *Engineer et al., 2004*; *Cai et al., 2009*; *Jakkamsetti et al., 2012*), and in whisker S1 of adults (*Polley et al., 2004*). But how enrichment affects maps at sub-columnar and cellular resolution is unknown. If the dispersed, salt-and-pepper whisker map in L2/3 is a byproduct of inadequate sensory (or non-sensory) experience, environmental enrichment should increase columnar structure and refine map topography at the cellular scale.

To test this question, we raised young adult mice from P21 for 25–50 days in either standard laboratory housing with a single littermate, or in an enriched environment with tactile toys and 1–2 additional littermates, which enhances whisker contact as part of social interactions (*Rao et al., 2014*). We then characterized whisker tuning and somatotopic map organization in L2/3 or L4 excitatory cells using 2-photon imaging of GCaMP6s expressed conditionally using layer-specific Cre lines. Enrichment (EN) strongly increased columnar organization and reduced salt-and-pepper organization, which increased the efficiency of population coding of whisker deflections within each column. These effects occurred mostly in L2/3, not L4. Thus, enrichment greatly enhances columnar organization in L2/3 of S1, so that cellular-scale functional topography better matches the anatomical columnar map.

## Results

### Whisker receptive fields measured by calcium imaging

We expressed GCaMP6s in either L2/3 pyramidal (PYR) cells or L4 excitatory cells of S1 by injecting Cre-dependent GCaMP6s virus into either Drd3-Cre mice for L2/3 PYR cell-specific expression or Scnn1a-Tg3-Cre mice for L4 excitatory-specific expression (*Gong et al., 2007*; *Madisen et al., 2010*; *Adesnik et al., 2012*). After several weeks for expression, we imaged neural activity in D2 and adjacent whisker columns, using resonant-scanned 2-photon calcium imaging through a chronic cranial window. Imaging was performed under anesthesia, while applying calibrated whisker deflections. We randomly interleaved deflection of 9 whiskers in a 3 × 3 grid centered on D2, using brief 5-pulse deflection trains, plus no-stimulation 'blank' trials (50–70 repetitions of each stimulus) (*Figure 1A–C*). Whisker-evoked ΔF/F responses were measured from regions of interest (ROI) corresponding to GCaMP6-expressing somata. ROIs were considered significantly whisker-responsive if the distribution of whisker-evoked ΔF/F (quantified in a 1 s period after whisker deflection) was greater than blank trials (permutation test, α = 0.05 corrected for nine whiskers, see Materials and methods). Analyses were restricted to whisker-responsive ROIs. Receptive fields of 3 example L2/3 cells in one column are shown in *Figure 1D*. Imaged neurons were localized post-hoc relative to column boundaries by histological staining and alignment by surface blood vessels (*Figure 1E*). We confirmed that GCaMP6s-based receptive fields were similar to spiking-based receptive fields by simultaneous juxtacellular spike recording and GCaMP6s imaging in 10 whisker-responsive L2/3 neurons. Receptive fields from spike counts were highly similar to those from ΔF/F in 9/10 cells (*Figure 1F*).

### Enrichment sharpens the point representation of a single whisker in L2/3

To study enrichment effects on the whisker map, littermates were separated into enriched (EN) and normally housed (NH) cohorts at weaning (P21). EN mice were housed with 4–5 tactile toys and 2–3 littermates, while NH mice were housed with a single littermate and no toys (*Figure 1A*, *Figure 1—figure supplement 1*). Toys consisted of large burrowing toys (tubes and huts that mice can enter),

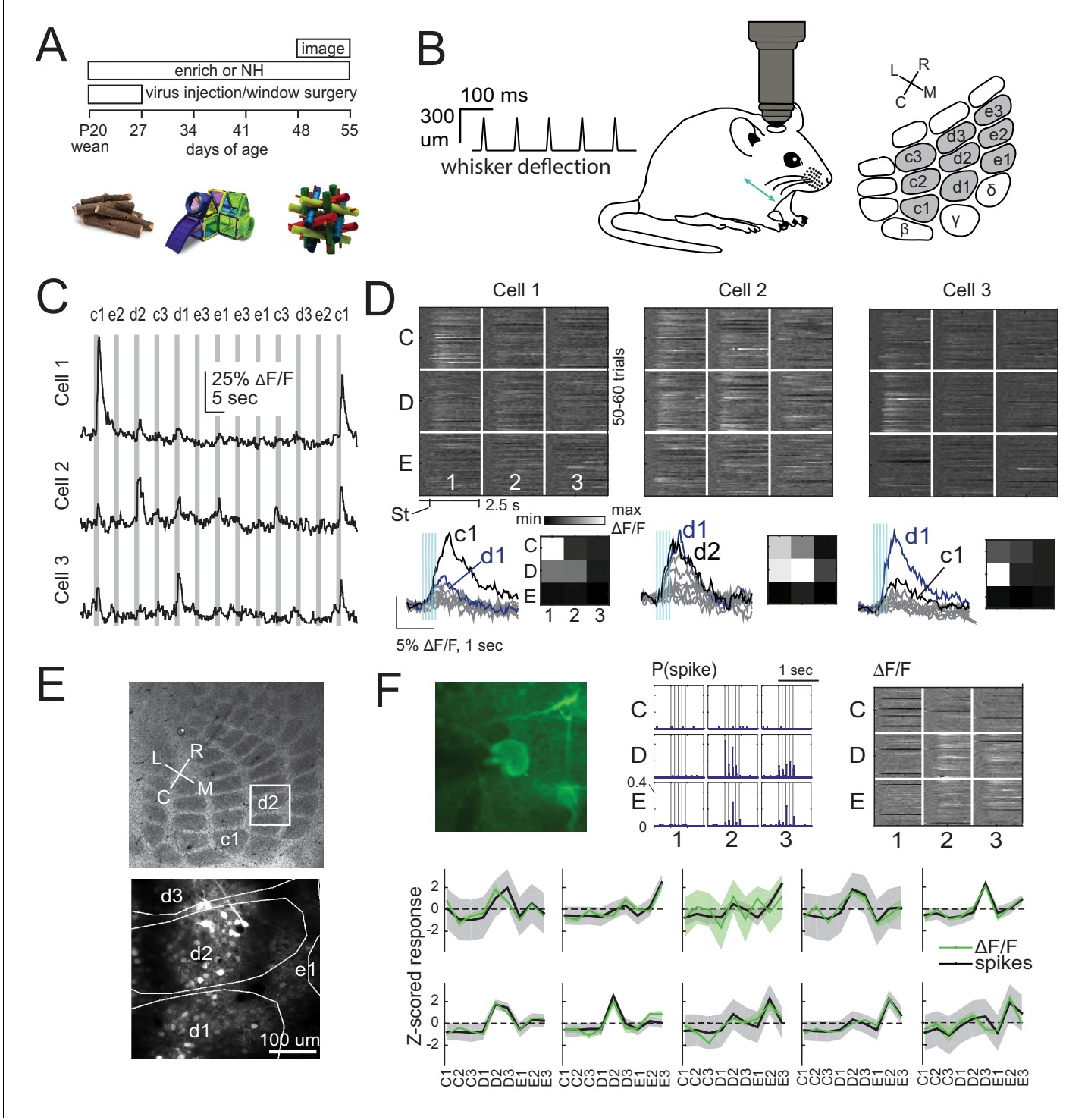

**Figure 1.** 2-photon calcium imaging of whisker responses and receptive fields in S1. (A) Experimental time line and example enrichment objects. (B) Schematic of whisker stimulus train and barrel map showing the 3 × 3 array of stimulated whiskers in gray. (C) Example ΔF/F traces from 3 ROIs imaged simultaneously in L2/3 of one column from a Drd3-Cre mouse. Gray bars are deflections of each indicated whisker. (D) Whisker receptive fields for the 3 cells in (C). Top, ΔF/F traces (shown as grayscale after baseline subtraction) for each trial for the nine whiskers (C1 to E3), aligned to stimulus onset (St). Bottom left, Mean ΔF/F trace for each whisker. Vertical bars are whisker deflections. The two strongest whiskers are shown in black and blue, respectively. Bottom right, Median ΔF/F averaged over 1 s response window. (E) Localization of one L2/3 imaging field (box) in a Drd3-Cre mouse relative to column boundaries in a cytochrome oxidase-stained section through L4 (top). Bottom, projection of barrel boundaries onto this imaging field. (F) Comparison of receptive fields measured by simultaneous GCaMP6s imaging and loose-seal cell-attached spike recording. Top: One example
*Figure 1 continued on next page*

*Figure 1 continued*

L2/3 neuron with its spiking receptive field shown as a peristimulus time histogram for each whisker (center), and its imaging receptive field from ΔF/F (right). Bottom: Average imaging and spiking receptive fields for each whisker-responsive neuron (n = 10). Shading is SEM.

DOI: https://doi.org/10.7554/eLife.46321.002

The following figure supplement is available for figure 1:

**Figure supplement 1.** Home-cage enrichment paradigm.

DOI: https://doi.org/10.7554/eLife.46321.003

medium-sized toys with complex shape and texture, and small wooden toys that could be moved by the mice (see Materials and methods). Mice interacted with the toys, which were replaced every 2–4 days to maximize novelty. This enriched environment will enhance not only tactile experience, but also social interaction, activity level, and other factors, all of which could drive cortical plasticity. Imaging was performed after 25–50 days of EN or NH experience (*Figure 1A*). The total number of mice and imaging fields are given in *Table 1*.

We first assessed enrichment effects on the point representation in S1, defined as the spatial profile of activity in S1 evoked by deflection of a single whisker. We quantified each cell's response to a reference whisker as evoked ΔF/F relative to spontaneous activity (blank trials), and then calculated the mean response across cells as a function of distance to the reference whisker's column center. This analysis only includes whisker-responsive cells. We first studied L2/3 PYR cells in Drd3-Cre mice. For L2/3 cells in normally-housed (NH) mice (11 imaging fields, seven mice, 998 total cells, 475 responsive cells, age: P56 ± 11), the point representation was centered on the reference whisker's anatomical column, and fell off gradually over several columns' distance. Enriched (EN) mice (11 imaging fields, seven mice, 1257 total cells, 874 responsive cells, age: P63 ± 16) showed an elevated mean response in the reference column and just around it. This was observed in the central bin (0–75 μm from column center; NH: 0.59 ± 0.06, EN: 0.77 ± 0.04; p=0.0098, permutation test for difference in means) as well as at 150–225 μm (NH: 0.53 ± 0.02, EN: 0.60 ± 0.02; p=0.0028) and 225–300 μm (NH: 0.41 ± 0.02, EN: 0.51 ± 0.01; p=0.0001). But EN suppressed mean responses at the flanks of the point representation (375–750 μm, p<0.05 for all bins) (*Figure 2A*). This enhancement of responses in the home column and reduced responses some distance away was also apparent in 2D maps in which imaging fields were rotated to align the arc axis, and then ROIs were spatially binned relative the reference column center (*Figure 2B*). Thus, enrichment sharpened the point representation in L2/3.

L4 is expected to contain a more spatially precise whisker map than L2/3 (*Simons, 1978*; *Andrew Hires et al., 2015*), but this has not been tested with cellular-resolution imaging. We imaged L4 cells in Scnna1-Cre mice. We imaged from 107 responsive cells in L4 of NH mice (six imaging fields, five mice, 322 total cells, age: P70 ± 16). We found that the point representation was sharper in L4 than L2/3 (*Figure 2C*). In EN mice, we imaged from 181 responsive cells in L4 (eight imaging fields, six mice, 586 total cells, age: P58 ± 5). We found that EN reduced mean whisker-evoked responses for cells in most bins <375 μm from the reference whisker's column center (p<0.05 for each bin), including at 0–75 μm (NH: 1.11 ± 0.13, EN: 0.86 ± 0.06, p=0.034) (*Figure 2C*). This was also observed in 2D maps around the reference column (*Figure 2D*). Thus, enrichment

**Table 1.** Mouse information.

|  | Age at imaging | sex | Days enriched | Days expressing GcaMP6s | N mice | N imaging fields |
|---|---|---|---|---|---|---|
| Drd3-Cre; NH | P56 ± 11 | 2F, 5M | n/a | 17 ± 2 | 7 | 11 |
| Drd3-Cre; EN | P59 ± 12 | 2F, 5M | 37 ± 13 | 20 ± 5 | 7 | 11 |
| Scnn1-Tg3-Cre; NH | P70 ± 16 | 2F, 3M | n/a | 39 ± 12 | 5 | 6 |
| Scnn1-Tg3-Cre; EN | P58 ± 5 | 5F, 1M | 35 ± 4 | 28 ± 5 | 6 | 8 |
| all Drd3-Cre | P58 ± 11 | 4F, 10M | 37 ± 13 | 18 ± 4 | 14 | 22 |
| all Scnn1-Tg3-Cre | P64 ± 14 | 7F, 4M | 35 ± 4 | 34 ± 11 | 11 | 14 |

DOI: https://doi.org/10.7554/eLife.46321.004

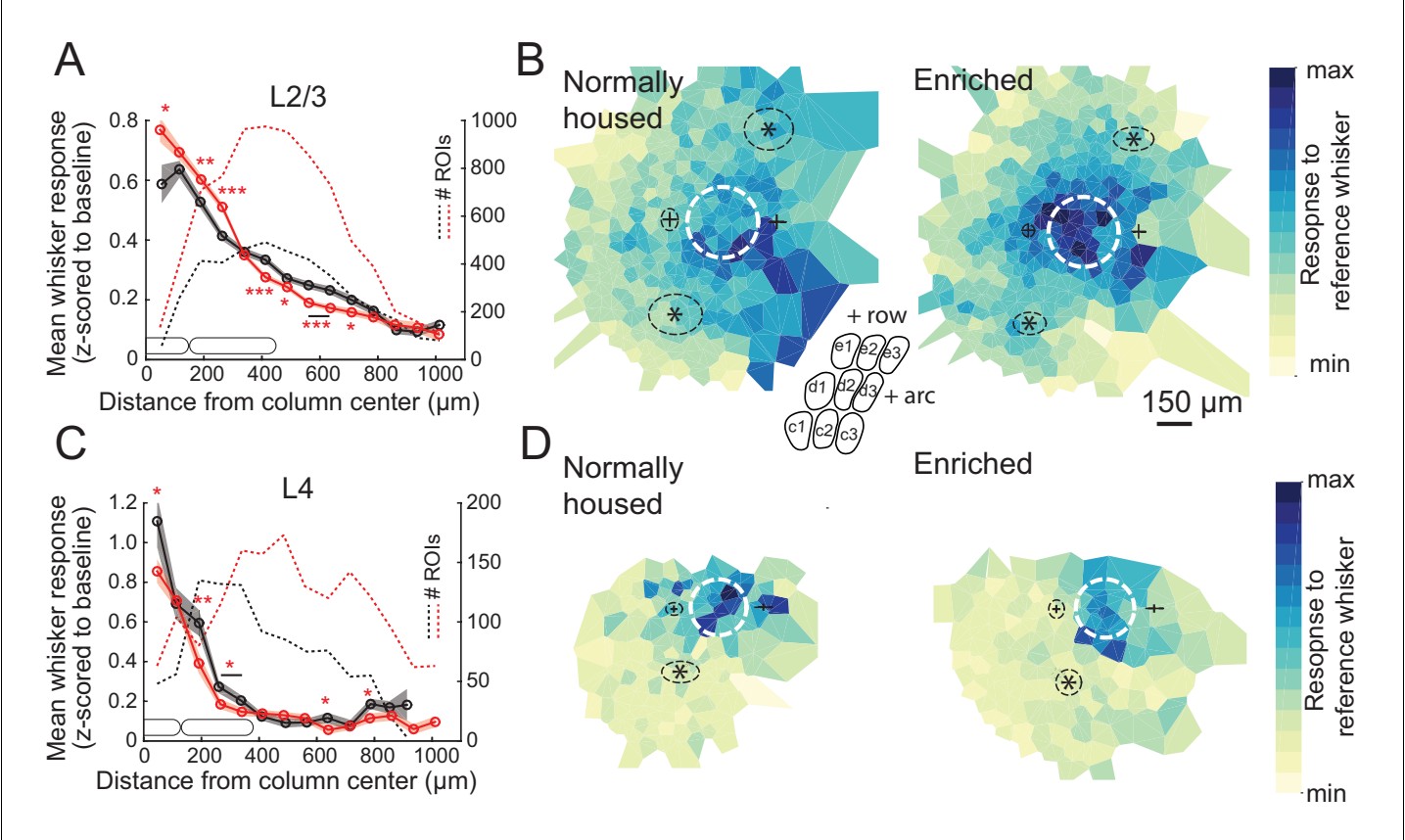

**Figure 2.** Enrichment strengthens and sharpens the whisker point representation in L2/3. (**A**) Average evoked response in L2/3 to a reference whisker (ΔF/F, z-scored to spontaneous activity in each cell) across all ROIs at increasing distances from the reference column center. ROIs are grouped into 75 μm width bins. Shading shows SEM. (**B**) Mean 2D spatial distribution of evoked responses to a reference whisker in L2/3. Dashed white circle is the reference whisker's column (shown as average barrel diameter around column center). All whisker-responsive ROIs within 750 μm of each reference column, across all imaging fields, are included. Color indicates mean response in each bin. Individual imaging fields were centered on the reference column and rotated to align the within-row anatomical axis horizontally. Pluses show mean location of column centers for within-arc, row+one and within-arc, row-1 whiskers. Stars show mean location of column centers for within-row, arc-1 and within-row, arc+one whiskers. Dashed black ellipses indicate standard deviation of these column centroids. Inset: schematic of barrel field in the same orientation as plots. (**C–D**) Same as A-B, but for L4 ROIs in Scnn1a-Cre mice.

DOI: https://doi.org/10.7554/eLife.46321.005

The following figure supplements are available for figure 2:

**Figure supplement 1.** Whisker point representation analyzed in a column-based reference frame.
DOI: https://doi.org/10.7554/eLife.46321.006
**Figure supplement 2.** Size of L4 barrels in enriched vs. normally housed mice.
DOI: https://doi.org/10.7554/eLife.46321.007

generally weakened L4 whisker responses, rather than sharpening the point representation relative to its central peak.

We also analyzed point representations as a function of neurons' columnar location, rather than as a function of absolute spatial distance to the reference column. To do this, the position of each ROI was determined in polar coordinates around the reference column center, in which angle was calculated relative to row-arc anatomical axes, and distance was expressed as a proportion of the reference column radius. ROIs were then spatially binned to calculate the point representation in this column-based reference frame. In L2/3, this confirmed that EN increased the mean response to a reference whisker in its home column, and revealed that EN sharpened the point representation primarily across whisker rows, not across arcs, and reduced spatial bias toward adjacent higher-arc whisker columns. In L4, this confirmed that EN weakened home-column responses (*Figure 2—figure supplement 1*).

To test whether enrichment also altered L4 barrel anatomical size, as occurs in rats (*Polley et al., 2004*), we quantified barrel radius for nine identified barrels from CO-stained tangential sections through L4 of each Drd3-Cre mouse. Enrichment did not alter mean barrel radius (NH: 151 ± 7.2 µm, n = 6 mice, EN: 145 ± 4.6 µm [mean ± SEM], n = 7 mice (2-way ANOVA, factors: enrichment and barrel identity, p=0.7 for enrichment factor) (*Figure 2—figure supplement 2*). There was also no change in barrel radius in Scnn1a-Cre mice, which have a slightly smaller barrels (NH: 118 ± 6.6 µm, n = 5 mice, EN: 124 ± 6.9 µm, n = 6 mice, p=0.01 for enrichment factor). Thus, the functional changes in L2/3 point representation occurred without gross changes in L4 barrel dimensions.

## Whisker map topography in L2/3 and L4 in normally housed mice

Because L2/3 exhibits pronounced salt-and-pepper tuning (*Sato et al., 2007*; *Clancy et al., 2015*; *Peron et al., 2015*), the sharpened point representation in L2/3 could reflect altered homogeneity of whisker tuning in each column. To test this, we first quantified salt-and-pepper organization in NH mice for L2/3 PYR cells (in Drd3-Cre mice) and L4 excitatory cells (in Scnna1-Cre mice). Prior studies of salt-and-pepper tuning did not distinguish between cell types in L2/3 (*Sato et al., 2007*; *Clancy et al., 2015*), and did not examine L4.

L2/3 PYR cells in NH mice had intermixed tuning for different whiskers in each individual column. We first identified the best whisker (BW) for each cell, defined as the whisker that evoked the absolute largest mean ΔF/F. Cells with different BWs were intermixed in each column (*Figure 3—figure supplement 1*). 53% (31/58) of responsive cells within 0–75 µm of a column center had a BW that matched the column's anatomically-mapped whisker (termed the columnar whisker, CW). The fraction of cells tuned to a reference whisker fell off gradually with cortical distance from that whisker's column center, to 16% (66/406 cells) at 225–300 µm and 1% (6/419 cells) at 525–600 µm (*Figure 3—figure supplement 1*). This distribution is broad compared to mean anatomical column radius (151 ± 7.2 µm). We also assessed tuning with a more rigorous approach that accounts for the statistical uncertainty in identifying the strongest whisker from limited stimulus repetitions. For this, we identified all whiskers that evoked a mean ΔF/F that was statistically indistinguishable from the BW (see Materials and methods). A cell was considered tuned for the CW if the CW was among these statistically equivalent best whiskers (eBWs). By this measure, 67% (39/58) of L2/3 PYR cells within 0–75 µm of the column center were tuned to the CW. This fraction fell off gradually with cortical distance, to 43% at 225–300 µm (174/406 cells) and 10% at 525–600 µm (41/419 cells) (*Figure 3—figure supplement 1*). This confirms salt-and-pepper tuning among L2/3 PYR cells in NH mice, and shows that the set of neurons tuned for a given whisker is dispersed across multiple columns (*Sato et al., 2007*; *Clancy et al., 2015*).

We applied the same analysis in L4 of NH mice. Surprisingly, while L4 showed a slight trend for more homogeneous tuning at the center of a whisker column (0–75 um) than L2/3, this was not significant. 63% of L4 excitatory cells (30/48) had a BW that matched the CW, and 73% (35/48) had the CW among the eBWs. These values are not different from L2/3 (p=0.43 and p=0.67 respectively, Fisher's Exact Test). However, the fraction of cells tuned to a given whisker fell off ~30% more sharply with distance in L4 than in L2/3, using either tuning measure. This difference was significant at distance bins from 150 to 375 µm from the column center (p<0.05, Fisher's Exact Test) (*Figure 3—figure supplement 1*). Thus, L4 exhibits more topographic precision in whisker tuning of single cells than L2/3.

## Enrichment decreases tuning heterogeneity in the salt-and-pepper map

Enrichment decreased local salt-and-pepper tuning heterogeneity in L2/3 in each column. For this analysis, we used the statistical definition of tuning in which a cell was considered tuned for a whisker if that whisker was among its statistically equivalent eBWs. In EN mice, 82% (111/135) of cells located 0–75 µm from a column center were tuned to the CW, which is greater than NH mice (67%, p=0.025, Fisher's Exact Test). This was true for all bins < 300 µm from the column center, but not further away (*Figure 3A*). When all cells located within column boundaries were considered together (irrespective of distance to center), 77.1% (468/607) of whisker-responsive cells in EN mice were tuned to the CW, compared to 65.1% (224/344) in NH (p=0.00008). Example imaging fields illustrating this effect are shown in *Figure 3B*. Thus, EN increased L2/3 map precision, particularly in the home column of each whisker, but did not abolish salt-and-pepper tuning (*Figure 3A*).

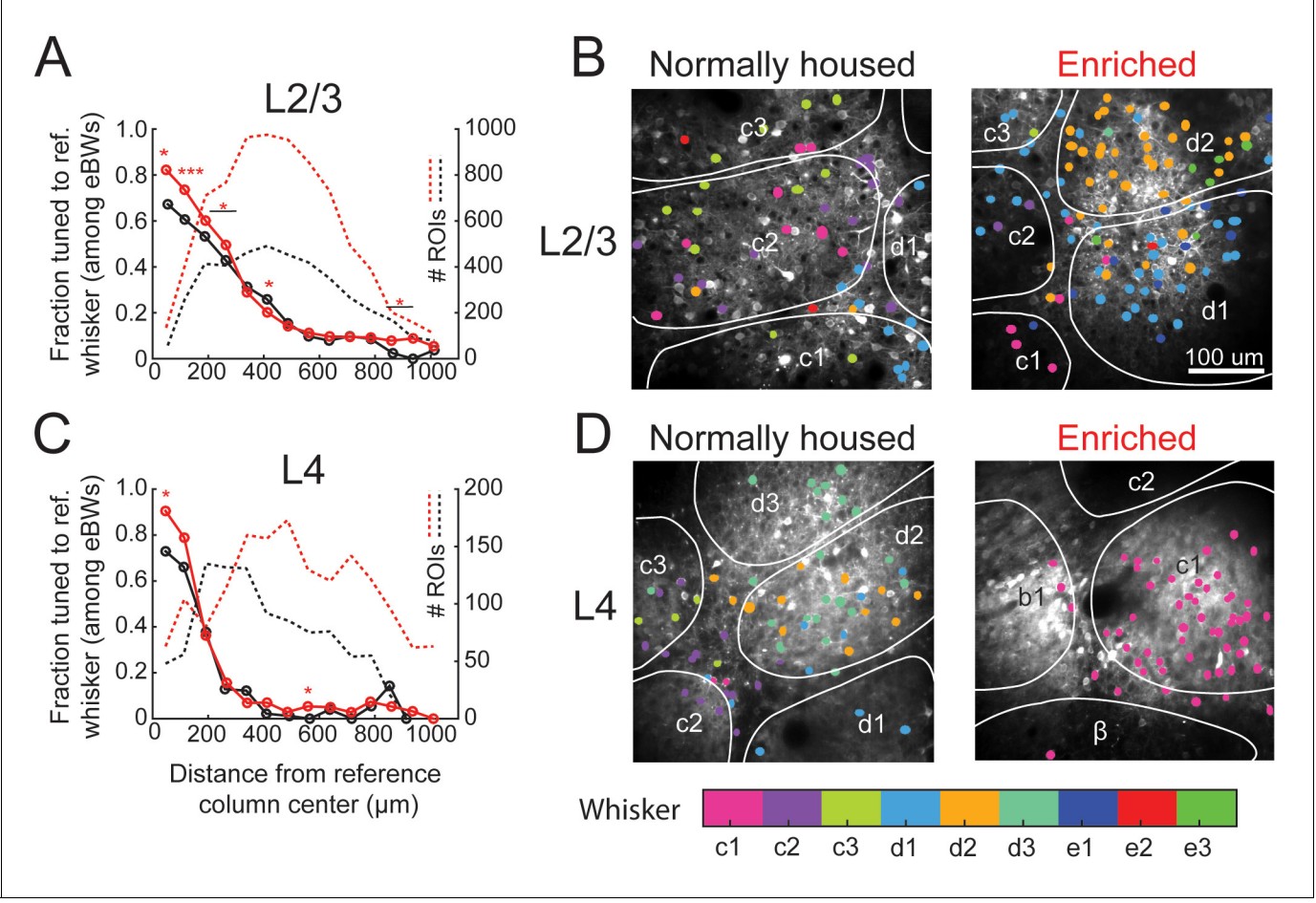

**Figure 3.** Effect of enrichment on somatotopic tuning precision. (**A**) Fraction of L2/3 PYR ROIs tuned to a reference whisker in EN and NH mice. Asterisks show significant differences between EN and NH by Fisher' Exact Test, computed separately in each spatial bin. *p<0.05, **p<0.01, ***p<0.001. A neuron was considered tuned for the whisker if that whisker was among the equal best whiskers (eBWs). (**B**) Example L2/3 imaging fields with neurons color-coded by whisker tuning. Left, Normally housed; right, Enriched. White lines denote barrel boundaries. Neurons located over septa and neurons located over barrels that were not tuned to the corresponding columnar whisker are color coded to the whisker which evoked the largest response magnitude. Neurons located over barrels are color coded to the columnar whisker if it evoked the largest response, or evoked a response that was statistically indistinguishable from the largest response. (**C–D**) As in A and B, for L4 excitatory neurons.

DOI: https://doi.org/10.7554/eLife.46321.008

The following figure supplement is available for figure 3:

**Figure supplement 1.** Salt-and-pepper tuning organization among L2/3 and L4 excitatory cells in normally housed mice.
DOI: https://doi.org/10.7554/eLife.46321.009

Enrichment also decreased tuning heterogeneity in L4 excitatory cells. In EN mice, 90% (57/63) of cells at column centers (0–75 μm) were tuned to the CW, which is greater than the 73% in NH mice (p=0.021) (*Figure 3C–D*). In EN mice, 89.3% (133/149) of all cells within a column boundary were CW-tuned, compared to 75.6% (65/86) in NH mice (p=0.0086). Thus, despite the expectation that L4 is less plastic than L2/3 (*Glazewski and Fox, 1996*; *Feldman and Brecht, 2005*), enrichment also increased tuning precision in L4.

## Enrichment sharpens whisker receptive fields

To test how enrichment altered whisker receptive fields of individual neurons, we first examined rank-ordered receptive fields, which quantify tuning sharpness independent of somatotopic shape. Whisker responses in each cell were normalized by Z-scoring to spontaneous (blank trial) activity (yielding a signal-to-noise measure of sensory responses), rank-ordered by response strength, and then averaged across cells. In L2/3, EN increased responses to the strongest three whiskers, and

decreased responses to the weakest three whiskers, relative to NH cells (*Figure 4A*). Thus, receptive fields were sharper, on average, in EN mice. EN specifically increased the signal-to-noise of both CW response and the best 2 SW responses (*Figure 4B*). In addition to these changes in tuning shape, EN also decreased spontaneous (blank trial) activity relative to NH (*Figure 4C*). Increased responses to the strongest whiskers and sharper tuning were also observed in absolute ΔF/F without normalization to spontaneous activity (*Figure 4—figure supplement 1*). Sharpening of tuning was more prominent for L2/3 cells located over L4 barrels than for those located over septa (*Figure 4— figure supplement 2*).

L2/3 receptive fields sharpened non-uniformly in somatotopic space. To quantify this, we compared responses evoked by the CW with those evoked in the same cell by row- and arc-surround whiskers (SWs) (*Figure 4D*). CW responses were higher in EN than NH (NH: 0.64 ± 0.03, EN: 0.71 ± 0.02; p=0.01, permutation test for difference in means). In NH, the SW one row dorsal to the CW (within-arc, −1 row) and one row ventral to the CW (within-arc, +one row) were 68% and 62% of the CW response, respectively. EN reduced responses to both these SWs (within-arc, −1 row: NH: 0.44 ± 0.02, EN 0.35 ± 0.02, p=0.001; within-arc, +one row: NH: 0.39 ± 0.02, EN: 0.29 ± 0.01, p=0.0001). As a result, these whiskers evoked only 49% and 41% of the CW response. This represents sharpening of the whisker receptive field across rows (*Figure 4D*). In the within-row dimension, EN increased responses to the SW one arc rostral of the CW (within-row, +one arc: NH: 0.34 ± 0.01, EN: 0.53 ± 0.02, p=0.0001), and did not alter responses to the SW one arc caudal of the CW (within-row, −1 arc: NH: 0.57 ± 0.04, EN 0.55 ± 0.04, p=0.40). Thus, EN broadened receptive fields in the within-row dimension (*Figure 4D*). These receptive field changes are consistent with the apparent changes in point representation in *Figure 2B*.

In L4, EN reduced responses to most whiskers in rank-ordered receptive fields (*Figure 4A*). Responses to the best surround whisker were reduced somewhat more than responses to the CW (*Figure 4A–B*), and spontaneous activity was unaffected (*Figure 4C*). The mean receptive field was sharper in L4 than L2/3 in both NH and EN, consistent with sharper point representation in L4 (*Figure 4A–B*). When receptive fields were examined in somatotopic space, CW responses were weaker in EN than NH (NH: 0.98 ± 0.09, EN: 0.81 ± 0.04, p=0.02), as were responses to the within-row, +one arc SW (NH: 0.54 ± 0.08, EN: 0.17 ± 0.02, p=0.0001), the within-row, −1 arc SW (NH: 0.67 ± 0.13, EN 0.33 ± 0.04, p=0.003), and the within-arc, +one row SW (NH: 0.23 ± 0.03, EN 0.11 ± 0.02, p=0.0007). The within-arc, −1 row SW was not sampled sufficiently in our L4 dataset to enable this comparison. Thus, EN sharpened L4 receptive fields, but by a global weakening of whisker responses, unlike in L2/3. This effect differs from adult enrichment in rats, which shrinks whisker receptive fields in L2/3 but not L4 (*Polley et al., 2004*).

## Enrichment strengthens functional column boundaries in L2/3

Prior imaging studies detected a global tuning gradient in L2/3 of S1 but no functional boundaries at anatomical column edges (*Kerr et al., 2007*; *Sato et al., 2007*). To test whether enrichment drives emergence of functional boundaries, we first examined tuning similarity (signal correlation) between pairs of simultaneously imaged L2/3 neurons in Drd3-Cre mice. In NH mice, mean signal correlation for neuron pairs fell off modestly with intersoma distance, and was slightly higher for within-column pairs than across-column pairs. For closely spaced pairs (<150 μm apart) signal correlation was only modestly higher for within-column pairs than across-column pairs (0.652 ± 0.005 vs. 0.611 ± 0.007, p<0.0001, permutation test of difference in means), reflecting the lack of functional column boundaries (*Figure 5A*). In EN mice, mean signal correlation for within-column pairs was modestly increased (NH: 0.637 ± 0.004, EN: 0.659 ± 0.002; includes pairs at all distances; p<0.0001, permutation test), while signal correlation for across-column pairs was greatly reduced (NH: 0.541 ± 0.004, EN: 0.425 ± 0.004; p<0.0001) (*Figure 5A,C*). Moreover, signal correlation for closely spaced cross-column pairs (<150 μm apart) was much lower than for similarly spaced within-column pairs (0.674 ± 0.003 vs. 0.529 ± 0.006, p<0.0001) (*Figure 5A*). Thus, EN generated a tuning boundary at column edges.

EN also generated columnar structure in noise correlations, which measure shared, stimulus-independent trial-to-trial variability between neurons, likely reflecting shared functional input (*Averbeck et al., 2006*; *Kohn et al., 2016*). In NH mice, noise correlations for L2/3 neuron pairs decreased with distance, and were similar within columns (0.334 ± 0.002) and across columns (0.327 ± 0.002). EN modestly reduced noise correlations for within-column pairs (to 0.307 ± 0.001;

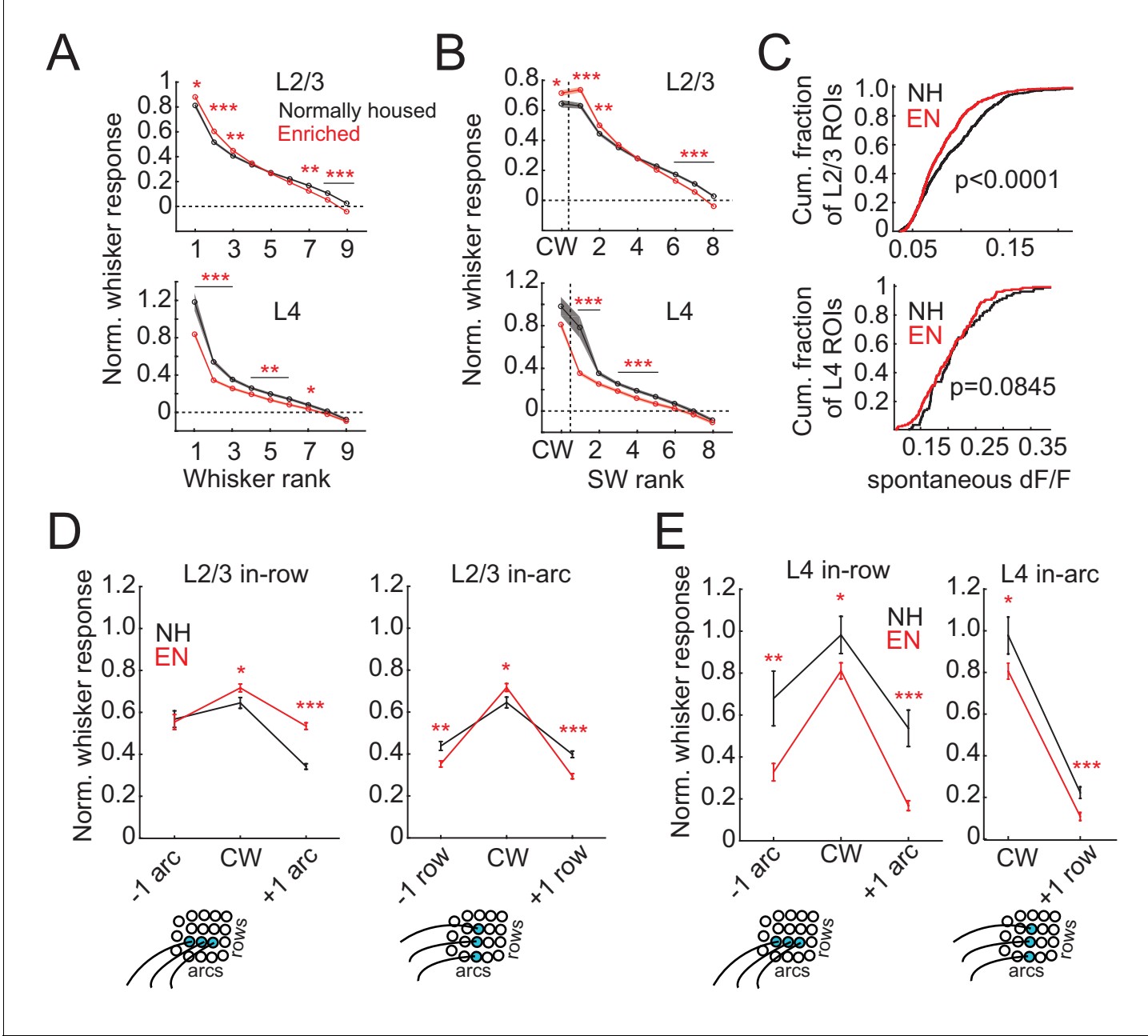

**Figure 4.** Enrichment sharpens whisker tuning curves. (**A**) Mean receptive field for all responsive ROIs, calculated after ranking whiskers from strongest to weakest in each cell and normalizing median responses to spontaneous activity. Asterisks, significant difference between EN and NH, computed separately by permutation test for each whisker rank. *p<0.05, **p<0.01, ***p<0.001. (**B**), Mean receptive fields separated into CW and ranked SW whiskers. (**C**) Effect of enrichment on spontaneous activity, defined as median ΔF/F during blank trials. P-values computed by permutation test. (**D**) Mean responses of L2/3 ROIs to CWs and adjacent whiskers within the same row (left) or the same arc (right). Lower panels: schematics of adjacent whiskers within rows or arcs on the whisker pad. (**E**) As in D, for L4 ROIs. Only columns containing at least 3 ROIs were included in this analysis. Because most ROIs were located in the C row, no data was available for responses to surround whiskers within the same arc but −1 row relative to the CW. Asterisks, significant difference between EN and NH. *p<0.05, **p<0.01, ***p<0.001.

DOI: https://doi.org/10.7554/eLife.46321.010

The following figure supplements are available for figure 4:

**Figure supplement 1.** Response magnitude and tuning sharpness analyzed without normalization to spontaneous activity.
DOI: https://doi.org/10.7554/eLife.46321.011

**Figure supplement 2.** Sharpening of tuning is more prominent for L2/3 neurons overlying L4 barrels than L4 septa.
DOI: https://doi.org/10.7554/eLife.46321.012

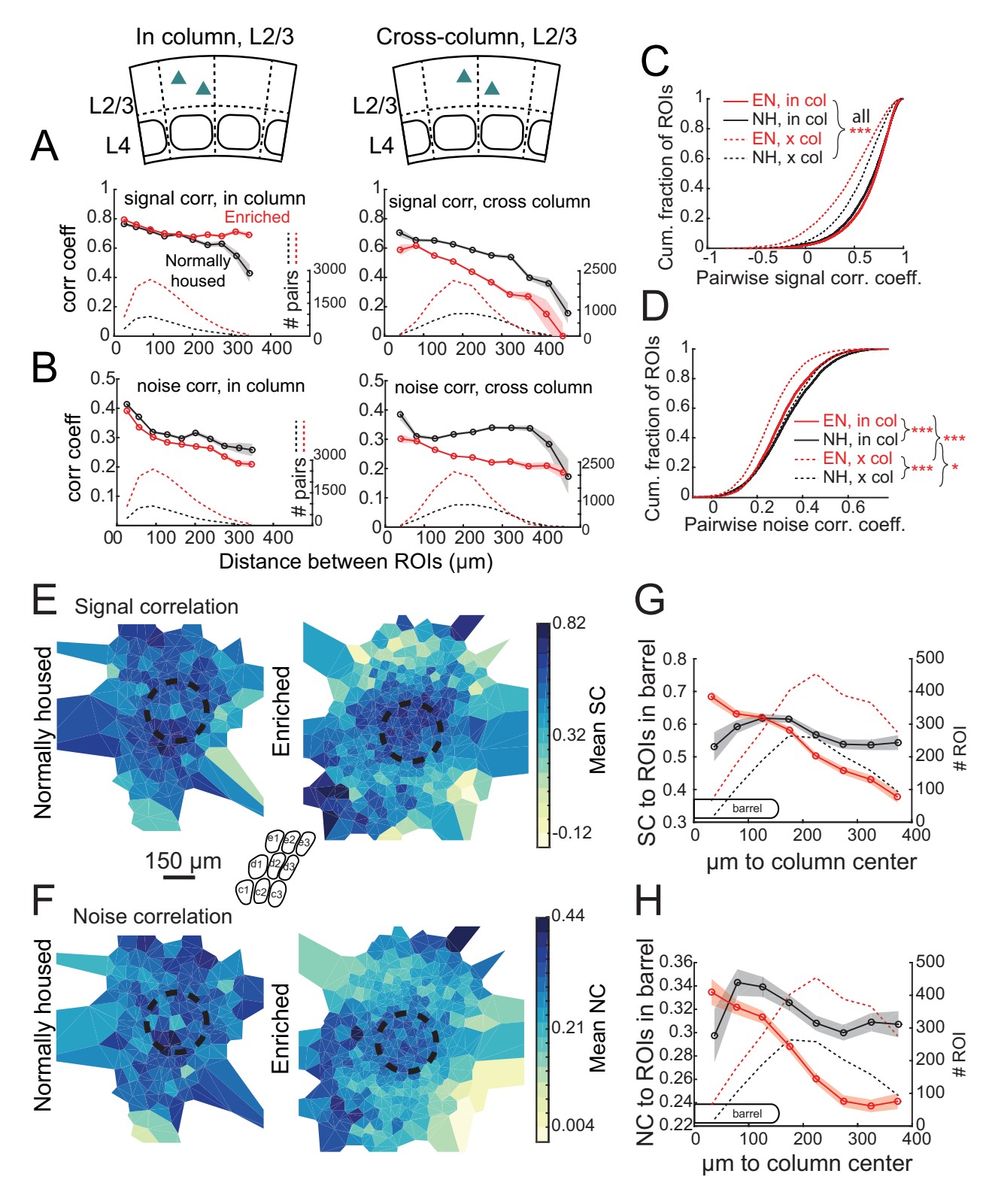

**Figure 5.** Enrichment alters the spatial structure of signal and noise correlations in L2/3. (**A**) Mean signal correlation across responsive L2/3 cell pairs as a function of inter-ROI distance, for cell pairs within a column (left) or across columns (right). Shaded regions denote SEM. Dashed, number of cell pairs in each bin. Insets show schematic of cell locations. (**B**) Mean noise correlation across all responsive L2/3 pairs, plotted as in (**A**). (**C–D**) Cumulative distribution of signal correlation (**C**) or noise correlation (**D**) for all within-column (in-col) and across-column (x-col) pairs. Asterisks denote difference

*Figure 5 continued on next page*

*Figure 5 continued*

between distributions computed by ANOVA with multiple comparisons correction. (E) Spatial map of mean signal correlation for sample ROIs within a position bin to all ROIs located within a reference whisker column. For each sample ROI, the mean signal correlation to all ROIs within the reference column was calculated. Sample ROIs were then clustered into spatial bins, and the mean signal correlation for each bin was plotted. The dashed circle is the reference column (shown as an average barrel diameter around the column center). (F) Spatial map of mean noise correlation to all ROIs within the reference whisker column, shown as in (E). (G) Mean signal correlation from sample ROIs to all cells in a reference column, as a function of sample ROI distance from column center. Shading is SEM. (H) Same as for (G), but for noise correlations. G-H show that enrichment steepens correlation gradients at column edges.

DOI: https://doi.org/10.7554/eLife.46321.013

The following figure supplements are available for figure 5:

**Figure supplement 1.** Spatial structure of signal and noise correlations across rows and arcs in L2/3.

DOI: https://doi.org/10.7554/eLife.46321.014

**Figure supplement 2.** Neuropil subtraction minimally impacts the major effects of the study.

DOI: https://doi.org/10.7554/eLife.46321.015

p<0.0001, permutation test vs. NH), but greatly reduced noise correlations for cross-column pairs (to 0.255 ± 0.001; p<0.0001, permutation test vs. NH). This effect was apparent even for cross-column pairs 100 µm apart (*Figure 5B,D*). The reduction of signal and noise correlations across column boundaries was apparent across both within-row and within-arc column boundaries (*Figure 5—figure supplement 1*).

To visualize functional columnar geometry based on these correlations, we calculated each neuron's mean signal and noise correlation to the population of neurons located within the anatomical column of a reference whisker. In NH, neurons outside the reference column exhibited fairly high signal and noise correlations to the neurons inside the column. EN substantially reduced these signal and noise correlations for neurons outside the column, while preserving signal correlation and only modestly reducing noise correlation among neurons within the reference column (*Figure 5E–F*). This generated a stronger gradient of signal- and noise-correlation in L2/3 at the reference column edge (*Figure 5G–H*).

In L4 of Scnn1a-Cre mice, EN generally increased correlations between co-columnar neurons (*Figure 6*). This was true for both signal correlations (NH: 0.681 ± 0.011; EN: 0.774 ± 0.005; p<0.0001) and noise correlations (NH: 0.241 ± 0.006; EN: 0.356 ± 0.002; p<0.0001). Too few cell pairs were imaged across columns to analyze cross-column correlations. Thus, EN drove functional differentiation of columns in L2/3, and increased coordinated activity within each column in L4.

## Robustness to neuropil contamination

GCaMP6s-based measures of tuning, tuning heterogeneity, and activity correlations could be contaminated by out-of-focus fluorescence from neuropil around each ROI. The extent of such neuropil contamination is difficult to measure empirically, and 2-photon imaging studies apply different methods and degrees of correction. Above, we applied no neuropil correction in L2/3, because Drd3-Cre mice had relatively low neuropil signal, reflecting GCaMP6s expression in only ~50% of L2/3 PYR cells. We did apply neuropil correction in L4 (at weight r = 0.3, see Materials and methods), because Scnn1a-Cre drove denser GCaMP6s expression, and imaging deeper in cortex required higher laser power, which could increase out-of-focus fluorescence above the imaging plane. To confirm that the major effects of the study were robust to different levels of neuropil correction, we applied neuropil subtraction in both L2/3 and L4 at three weights (r = 0, r = 0.3, r = 0.7, see Materials and methods) (*Figure 5—figure supplement 2*). Enrichment effects on tuning heterogeneity (*Figure 3*) and point representation (*Figure 5*) were qualitatively unchanged across these levels of neuropil subtraction in both L2/3 and L4 (*Figure 5—figure supplement 2A–D*). The highest level of subtraction (r = 0.7) has been used in several prior studies (*Chen et al., 2013*) but here appeared to cause overcorrection for most ROIs in both L2/3 and L4, strongly reducing the number of ROIs that were whisker-responsive. Effects of enrichment on signal correlations were conserved for r = 0 and r = 0.3, but were lost for r = 0.7, likely because fewer cells could be analyzed (only responsive cells were analyzed). Effects of enrichment on noise correlations persisted across all neuropil subtraction levels (*Figure 5—figure supplement 2E–F*). Thus, the major effects of enrichment were robust to methodological choices on neuropil subtraction.

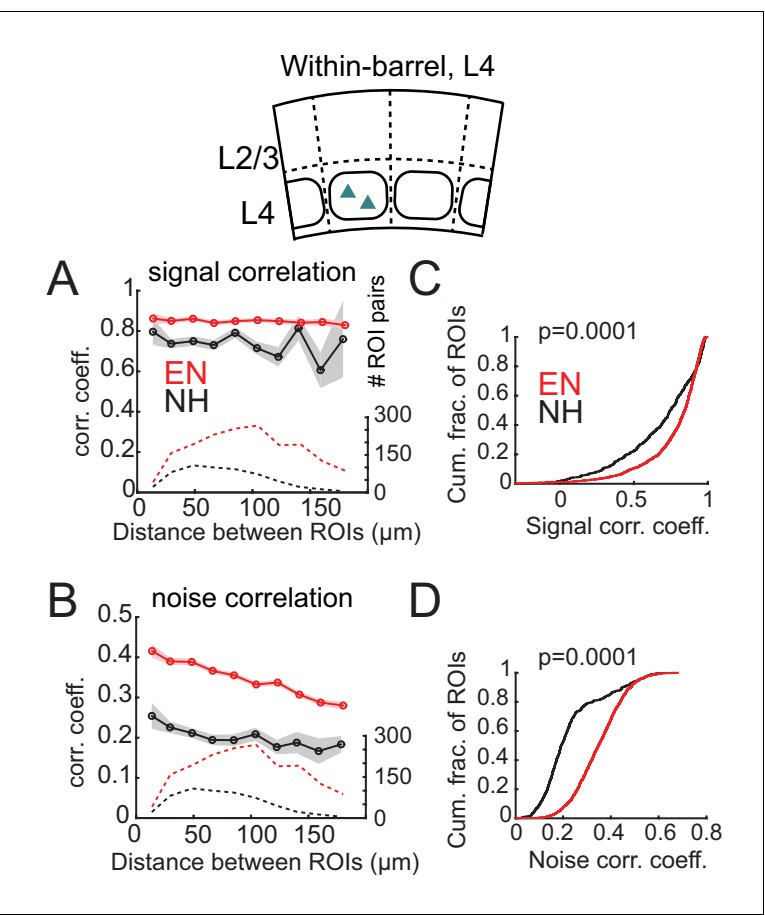

**Figure 6.** Enrichment increases signal and noise correlations within L4 barrels. (A) Mean signal correlation across responsive L4 cell pairs as a function of inter-ROI distance, for cell pairs within a column. Conventions as in *Figure 6A*. Inset, schematic of cell locations. (B) Mean noise correlation across within-column pairs, plotted as in (A). (C–D) Cumulative distribution of signal correlations (C) or noise correlations (D) for within-barrel ROI pairs in L4.

DOI: https://doi.org/10.7554/eLife.46321.016

## Enrichment improves population coding of columnar whisker deflections

Finally, we tested whether these changes in L2/3 sensory representation are sufficient to improve population coding of whisker input within local regions of S1. To do so, they must be large relative to the substantial single-trial variability that exists in S1 (*Clancy et al., 2015*). We built a neural population decoder that predicted CW deflection (relative to spontaneous activity) from single-trial ΔF/F from populations of imaged L2/3 neurons in Drd3-Cre mice. Each ROI was trained by logistic regression, on a subset of CW deflection and blank trials, to predict whisker deflection from ΔF/F in a 1 s window. Performance was tested on held-out trials. Single-ROI decoders performed above chance, with slightly better performance by EN cells than NH cells (fraction correct for NH: 0.63 ± 0.005, for EN: 0.65 ± 0.004, p=0.004, permutation test for difference in means). Population decoders were built by randomly selecting ensembles of 2–30 ROIs that were simultaneously imaged in the same field, and averaging the output of each ROI classifier to generate a population prediction on each trial. Because ROIs could be located in different whisker columns, we averaged performance across the set of CWs for those columns. Detection performance increased with population size, and was greater for EN than NH for populations containing ≥3 neurons (p<0.001; permutation test of difference in means) (*Figure 7A–C*).

To test how detection performance varied with columnar topography, we built decoders from spatially clustered neural ensembles (N=2-42 neurons) centered at varying distances from a

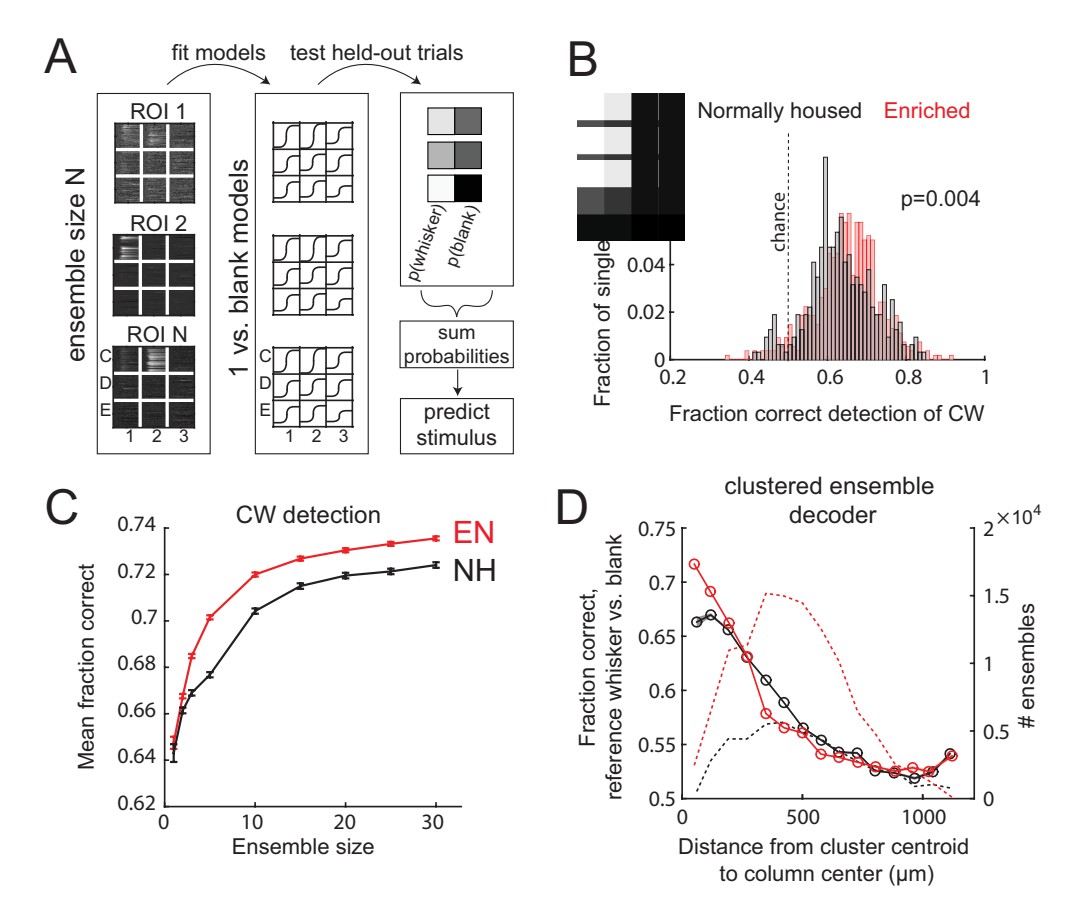

**Figure 7.** Enrichment improves population coding of CW deflections. (**A**) Design of neural decoder for detecting a whisker deflection. (**B**) Performance of single-ROI decoders on detection of CW deflection vs. blank (no whisker deflection). (**C**) Detection performance for population decoders of varying size. Each ensemble decoder consisted of randomly positioned, simultaneously imaged ROIs, and was tested on detection of the whisker closest to the centroid of the ensemble. Bars show SEM. (**D**) Detection of a reference whisker deflection by ensemble decoders located at varying distances from the reference column. These were spatially clustered ensembles of 2–42 simultaneously imaged ROIs (see Materials and methods). Distance bins refer to distance between reference column center and the centroid of each ensemble.

DOI: https://doi.org/10.7554/eLife.46321.017

reference whisker's anatomical column. Spatially clustered (not randomly dispersed) ensembles were selected in each imaging field (see Materials and methods). These decoders were tested for detection of reference whisker deflection versus a blank. Mean detection performance (on average across the population sizes) was greatest for ensembles centered near the whisker column center, and fell off with distance. EN decoders outperformed NH decoders for ensembles centered at 0-77 μm and 77-154 μm from a column center ($p=2*10^{-6}$, 2-way ANOVA), which are within an average column radius (**Fig. 7D**). This effect disappeared and even reversed for more distant ensembles, where EN performed worse than NH (e.g., 308-385 μm bin; $p=2*10^{-6}$). Thus, the strengthening and sharpening of the point representation in EN mice improves single-trial population detection of whisker deflections inside each whisker's column, and topographically sharpens effective decoding in S1.

## Discussion

### Whisker map structure in L2/3 and L4 of normally housed mice

Our findings confirm that S1 of NH mice exhibits dispersed, salt-and-pepper tuning organization in L2/3, without functional borders at column boundaries (*Sato et al., 2007*; *Clancy et al., 2015*). The

point representation of a whisker among L2/3 cells spread across multiple columns, and fell off gradually without discrete columnar boundaries (*Figure 2*). This was due in part to tuning heterogeneity within each column: only 48% of PYR cells in each column responded maximally to the CW, and only 65% had the CW among statistically equal best whiskers (*Figure 3—figure supplement 1*). L4 contained a more spatially focused map, as expected from thalamocortical topography and prior studies of L4 receptive fields (*Simons, 1978*; *Armstrong-James and Fox, 1987*; *Andrew Hires et al., 2015*), but still a small degree of salt-and-pepper structure (*Figures 2* and *3*). Thus, whisker somatotopy is more precise in L4 but more dispersed in L2/3, similar to tonotopy in rodent A1 (*Winkowski and Kanold, 2013*). This comparison between L4 and L2/3 assumes that Drd3-Cre and Scnn1a-Cre mice have overall similar somatotopic map organization.

GCaMP6- and spike-based tuning measurements will differ because of lower single-spike sensitivity with GCaMP, and the use of train stimuli rather than single whisker deflections. In addition, some non-columnar whisker responses in GCaMP imaging may reflect slow tangential activity propagation across columns (*Petersen et al., 2003*) within the 1 s response window. Despite these caveats, we found generally good correspondence between GCaMP6s- and spike-based tuning in calibration experiments (*Figure 1F*). Imaging is essential to measure map topography among PYR cells, because dense, cell-type specific recording with accurate soma localization is generally not possible using spike recordings.

## Enrichment increases receptive field and map precision in L2/3

While deprivation-induced plasticity has been well studied in S1 (*Feldman and Brecht, 2005*), the effects of enrichment are less understood. Enrichment can weaken and spatially sharpen whisker responses (*Polley et al., 2004*), or promote larger, more overlapping whisker responses (*Guic et al., 2008*), and can create smoother maps of cutaneous forepaw input (*Coq and Xerri, 1998*). These prior studies lacked single-cell mapping resolution and cell type specificity, and thus whether enrichment would transform the dispersed whisker map into a sharper map with more distinct columnar boundaries was unclear.

We found that enrichment strengthened and sharpened point representations among L2/3 PYR cells, by increasing whisker responses within and just around the home column, and decreasing responses for cells further away (*Figure 2*). Point representations were not sharpened isotropically, but instead preferentially sharpened in the across-row dimension (*Figure 2—figure supplement 1*). Similar results were observed for single-cell receptive fields: enrichment reduced salt-and-pepper tuning heterogeneity in column centers, strengthened CW responses, and sharpened whisker receptive fields, particularly in the across-row dimension (*Figures 3–4*). In L4, enrichment more subtly sharpened the whisker map, via general weakening of whisker responses (*Figure 2*). Because surround whisker responses weakened more than CW responses, L4 receptive fields sharpened around the CW and a greater fraction of neurons became tuned for the CW (*Figures 3–4*).

Thus, EN altered both L2/3, which is well known for robust experience-dependent plasticity, and L4, which retains some plasticity in adults (*Landers et al., 2011*; *Oberlaender et al., 2012*). In L2/3, map changes could reflect use-dependent strengthening of L4-L2/3 or local recurrent projections (*Clem and Barth, 2006*; *Ko et al., 2013*), reorganization of cross-columnar projections (*Finnerty et al., 1999*), or changes in POm thalamic (*Audette et al., 2019*) or top-down input (*LeMessurier and Feldman, 2018*) to L2/3.

Several prior studies found that enrichment enhances whisker-evoked spiking and local field potential responses in whisker S1, but did not measure effects on receptive fields or maps (*Alwis and Rajan, 2014*; *Devonshire et al., 2010*; *Megevand et al., 2009*). This enhanced magnitude of whisker responses echoes prior findings in auditory and visual cortex (*Alwis and Rajan, 2014*), matches our findings in L2/3 (*Figures 2* and *4*), and may involve spine formation induced by enrichment in S1 (*Landers et al., 2011*). One study used intrinsic signal imaging to discover sharpened whisker point representations in L2/3 after adult enrichment, caused by reduction of whisker response magnitude and sharpening of L2/3 single-unit receptive fields (*Polley et al., 2004*). Our cellular-resolution mapping substantially extends this finding, by showing that juvenile enrichment strengthens CW responses, sharpens the L2/3 whisker map, and generates functional tuning boundaries in L2/3 aligned to anatomical column edges. By imaging in L4, we detected modest receptive field plasticity, which was not detected in L4 multiunit recordings (*Polley et al., 2004*). These differences could also reflect enrichment age or species (mice vs. rats). Layer-specific population Ca

imaging allowed us to observe map sharpening among diversely tuned neurons, and to discover that enrichment regulates the degree of salt-and-pepper organization in S1.

## Enrichment enhances functional column boundaries in L2/3

S1 contains whisker-specific thalamocortical termination zones in L4 (*Woolsey and Van der Loos, 1970*) and strong radial L4-L2/3 connectivity (*Bender et al., 2003*; *Bureau et al., 2006*). Thus, it has been puzzling that prior cellular-resolution imaging (in standard-housed mice) revealed gradual somatotopic tuning gradients in L2/3 without major transitions in tuning at column edges (*Kerr et al., 2007*; *Sato et al., 2007*; *Clancy et al., 2015*). We hypothesized that appropriate experience could drive development of column-related functional organization in L2/3.

Functional organization at column boundaries was quantified using signal correlations, which reflect shared information coding, and noise correlations, which are thought to reflect membership in common functional networks (*Averbeck et al., 2006*; *Kohn et al., 2016*). In NH mice, signal and noise correlations fell off gradually with distance between neurons, as is typical for cortical circuits (*Montijn et al., 2016*; *Rose et al., 2016*). The spatial profile of correlations was similar within and across anatomical column boundaries, confirming gradual somatotopy and lack of sharp functional column borders (*Kerr et al., 2007*; *Sato et al., 2007*; *Clancy et al., 2015*). EN mice had modestly increased signal correlations within columns but substantially decreased signal correlation across columns, even for neurons located just ~100 µm apart across a column edge. Similarly, noise correlations were modestly decreased within columns, but strongly decreased across columns, even for neurons ~100 µm apart (*Figure 6*). Thus, enrichment enhanced tuning distinctions between columns, reduced shared noise between columns, and created sharper functional boundaries at column edges, evident from both signal and noise correlations. Local tuning heterogeneity still existed in L2/3 of EN mice, but functional columnar organization was stronger.

Greater differentiation of somatotopic tuning across columns could improve neural coding for whisker stimulus detection in each column. In addition, because noise correlations are often detrimental for sensory coding (*Kohn et al., 2016*), their reduction may improve whisker sensory coding for downstream areas. Reduced signal and noise correlations across columns in L2/3 may reflect more rigid columnar segregation of L4-L2/3 and L2/3-L2/3 circuits, for example by reducing cross-columnar excitatory projections or enhancing cross-columnar inhibition. In L4, enrichment increased signal correlations within columns, which may reflect stronger recurrent excitation in each L4 barrel (*Ashby and Isaac, 2011*).

## Enrichment improves population coding of whisker deflection in L2/3

Sensory tuning and maps reflect average responses over many stimulus presentations, but perception and sensory-evoked behavioral responses occur on single trials. Single L2/3 neurons in S1 exhibit high trial-to-trial response variability (*de Kock et al., 2007*; *Barth and Poulet, 2012*), which limits coding efficiency. To test whether sharper maps predict improved neural coding on single trials, we constructed neural decoders that predicted the presence of a whisker stimulus from measured single-trial responses of S1 neurons. Our focus was on coding by L2/3 PYR cell ensembles, which relay information from S1 to downstream cortical areas (*Chen et al., 2015*).

Enrichment improved detection of CW deflections by spatially clustered PYR ensembles located in column centers (*Figure 7*). Enrichment increased both absolute detection accuracy and coding efficiency such that in EN mice, smaller ensembles were needed for equivalent detection performance. Thus, the strengthening of CW-evoked responses, the reduction of spontaneous activity and noise correlations, and the higher proportion of CW-tuned neurons are sufficient to improve single-trial coding of whisker deflections in EN mice. Topographically, enrichment improved detection by ensembles in the home whisker column, but worsened detection by ensembles in neighboring columns (*Figure 7D*). Thus, the columnar organization of effective decoding was sharpened, matching the sharper point representations in L2/3 of enriched S1. These results do not predict whether behavioral detection of whisker input is altered by enrichment, but they do suggest that coding within individual S1 columns is more efficient.

## Dispersed, weakly columnar maps partially reflect impoverished experience

These results indicate that the dispersed, salt-and-pepper, weakly columnar somatotopy in L2/3 of S1 is partially a product of impoverished experience in standard laboratory housing. While enrichment promotes more diverse tactile experience, it also likely increases social interactions and affects general activity levels. Which of these factors is dominant in driving whisker map plasticity was not tested. Enrichment strengthened and sharpened the L2/3 PYR cell map and drove emergence of functional boundaries aligned to columnar structure. Enrichment reduced, but did not eliminate, salt-and-pepper tuning heterogeneity in L2/3. Thus, salt-and-pepper organization is a robust feature of rodent L2/3 sensory maps, but its degree is flexible based on experience. Enrichment reduced both mean tuning similarity and stimulus-independent (spontaneous) firing correlations across columns. Thus, columns became more functionally independent, which predicts more accurate column-based encoding of whisker input. The circuit mechanisms for emergence of columnar structure in L2/3 remain to be identified.

# Materials and methods

**Key resources table**

| Reagent type (species) or resource | Designation | Source or reference | Identifiers | Additonal information |
|---|---|---|---|---|
| Genetic reagent (*M. musculus*) | Scnn1a-tg3-Cre line | Jackson Labs | RRID:IMSR_JAX:009613 | |
| Genetic reagent (*M. musculus*) | Drd3-Cre line | MMRRC | RRID:MMRRC_034610-UCD | |
| Recombinant DNA reagent | AAV1.Syn.Flex. GCaMP6s.WPRE.SV40 | Penn Vector Core | Penn: AV-1-PV2821 | |
| Recombinant DNA reagent | AAV9.CAG.Flex. GCaMP6s.WPRE.SV40 | Penn Vector Core | Penn: AV-9-PV2818 | |
| Software, algorithm | MATLAB (version 2014b) | Mathworks | RRID:SCR_001622 | |
| Software, algorithm | ScanImage | Vidrio Technologies | RRID:SCR_014307 | |
| Software, algorithm | Igor Pro (version 6) | Wave Metrics | RRID:SCR_000325 | |

## Animals

All procedures were approved by the UC Berkeley Animal Care and Use Committee and follow NIH guidelines. For population imaging experiments we used 14 Drd3-Cre mice (10 males and four females), and 11 Scnn1a-Tg3-Cre mice (4 males and seven females) (*Table 1*). Drd3-Cre mice were obtained from MMRRC on a mixed C57BL6/FVB background (#034610-UCD), and were paired with C57BL/6 mice for several generations before generating the mice used in the current study. Scnn1a-Tg3-Cre mice were obtained on a C57BL/6 background from Jackson Laboratories (JAX #009613) and were maintained on a C57BL/6 background. Juxtacellular recordings were performed in 5 C57BL/6 mice (Jackson Laboratories). Whether the Drd3-Cre and Scnn1a-Cre transgenes themselves impact somatotopic map organization or plasticity is not known. But salt-and-pepper organization was previously observed in L2/3 in wild-type C57BL/6 mice (*Clancy et al., 2015*), so the presence of the Drd3-Cre transgene does not appear to strongly impact basic map organization.

## Enrichment

Littermates were separated into normally housed (NH) and enriched (EN) groups at weaning (P20-21). EN animals were housed with 2–3 littermates in standard mouse cages (30 × 14 × 18 cm) into which a nesting enclosure and several toys were added and exchanged every 2–4 days. Toys included sticks and blocks made from wood, ridged plastic, PVC, rubber, or cement, chosen for diversity of shape and texture (*Figure 1A*, *Figure 1—figure supplement 1–*). Toys were selected so that at all times EN cages contained one burrowing toy, one medium sized toy, and several small

wooden toys that could easily be moved by the mice. Toys were cleaned and sterilized between uses. NH mice were housed in the same cage, with one littermate and no toys. All cages contained bedding and nesting material. Running wheels were not used in either NH or EN cages.

## Surgery

A single surgery for headgear implant and viral injection was performed at ~P41 (Drd3-Cre mice) or ~P30 (Scnn1a-Cre mice). Before surgery, mice received dexamethasone (2 mg/kg, i.p.), meloxicam (10 mg/kg) and enrofloxican (5 mg/kg) for analgesia and to prevent infection. Mice were anesthetized with isoflurane and maintained at 37°C. The skull was cleaned and a stainless steel head-holder containing a 5 mm aperture was affixed to the skull with dental cement. Intrinsic signal imaging (ISOI) was performed through the intact skull to localize D1, D2 and D3 columns in S1 (*Drew and Feldman, 2009*), and a 3 mm craniotomy was made over the D2 column. A Cre-dependent GCaMP viral vector was injected at three nearby sites, at two depths each. For L2/3 (Drd3-Cre mice), we used AAV1-Syn-Flex-GCaMP6S-WPRE, injected at ~200 and ~300 μm below the dura. For L4 (Scnn1a-Cre mice), we used AAV9-CAG-Flex-GCaMP6s-WPRE, injected ~350 and ~450 μm below the dura. Viruses were obtained from UPenn Viral Vector Core via the Janelia GENIE Project (*Chen et al., 2013*). The craniotomy was sealed with a #1 glass coverslip (0.15 ± 0.02 mm thick, 3 mm diameter) cemented to the skull. Mice received post-operative buprenorphine (0.1 mg/kg, i.p.) for analgesia.

Following implant surgery, mice were kept in a warm environment until they fully recovered posture and locomotion (<1 hr), and were then returned to their home cage. At least two mice from the same home cage underwent surgery on the same day, so they could recover in sync. Thus, in each NH cage, both mice were recovering from surgery at the same time, and in each EN cage, at least 2 out of the 3–4 mice were recovering at the same time.

Surgical recovery was quantified in a special cohort of 6 NH and 6 EN mice (separate from the mice used in imaging). Mice recovered to pre-surgical weight within 4 days (NH) or 5.5 days (EN), on average. General exploratory behavior was measured daily from 1 to 7 days post-surgery by placing each mouse in a clean cage with bedding but no toys or nesting material. After a 10 min habituation period, we quantified time spent in three spontaneous behaviors (locomoting, grooming, rearing) for a 10 min observation period. The prevalence of these behaviors did not differ between EN and NH mice, although EN mice showed a non-significant trend for more locomotor behavior (data not shown). Thus, general exploratory behavior during surgical recovery was roughly similar between NH and EN mice. Behavior in the home cage was not measured because enrichment objects and cage top obscured observation. Thus, surgical recovery was relatively fast and similar between NH and EN groups, and was a small fraction of total 35–40 day NH or EN duration before imaging (*Table 1*).

## 2-photon calcium imaging

Imaging took place 18 ± 4 days after virus injection for Drd3-Cre mice (*Gong et al., 2007*) and 34 ± 11 days after virus injection for Scnn1-Tg3-Cre mice (*Madisen et al., 2010*; *Adesnik et al., 2012*), as required for adequate GCaMP expression in L2/3 and L4 (*Table 1*). Imaging was performed once daily observation revealed stable numbers of fluorescent neurons with few or no neurons exhibiting nuclear expression (*Tian et al., 2009*). 2-photon imaging was performed using a Sutter Moveable-Objective Microscope with one resonant and one galvo scan mirror. We used a 16x, 0.8 NA water-dipping objective (Nikon). Excitation was delivered with a Coherent Chameleon Ti-Sapphire pulsed laser tuned to 920 nm. Fluorescence emission was filtered with a Chroma HQ 575/50 filter and detected with a Hamamatsu photomultiplier tube (H10770PA-40). Single Z-plane images (512 × 512 pixels) were collected at 30 Hz frame rate using ScanImage (*Pologruto et al., 2003*).

Imaging was performed under low-dose isoflurane (<1% in oxygen) combined with the sedative chlorprothixene (0.08 mg, i.p.), with body temperature stabilized at 37°C. L2/3 imaging fields were ~400 × 400 μm, located ~100–250 μm below the dura, and contained ~50–100 expressing neurons. In Scnn1a-Cre mice, smaller imaging fields were often used (~300 × 300 μm) to compensate signal-to-noise ratio for imaging depth.

nine whiskers in a 3 × 3 array centered on D2 were inserted into nine independent, calibrated piezoelectric actuators, positioned ~5 mm from the face. A plastic shield kept away nearby, non-

stimulated whiskers. Whisker stimuli were presented as trains (five impulses, 100 ms apart) of individual deflections (300 µm rostrocaudal deflection, rise/fall time 4 ms, total duration 10 ms). Trains are necessary due to sparse spiking in L2/3 (*de Kock et al., 2007*; *Barth and Poulet, 2012*), and increase the likelihood of eliciting detectable GCaMP signals. We interleaved the nine whisker stimuli (5 s isi, random order) plus 'blank' trials in which no whisker was deflected. At each imaging field, $60 \pm 10$ repetitions of each whisker stimulus were presented, and imaging data was collected as approximately forty 80 s long movies. 1-2 fields were imaged in each mouse, several days apart.

After imaging was complete for all co-housed mice within a given cage, all mice in that cage were euthanized. The brain was removed and fixed in 4% paraformaldehyde for several days before the cortex was flattened and sectioned at 50 µm parallel to the cortical surface. Sections were processed for cytochrome oxidase to reveal the barrels in L4. Imaged neurons were localized relative to barrel column boundaries by alignment using surface blood vessels and Z-stacks of GCaMP expression collected during the imaging sessions, as well as by fiducial marks at the pial surface created by high-power laser scanning after imaging was complete. We used a custom MATLAB program to localize the centroids of the nine anatomical barrels that corresponded to the stimulated whiskers, and to calculate the X-Y coordinate of each ROI relative to these centroids.

Genetic expression of GCaMP is unlikely to affect basic L2/3 map development, because salt-and-pepper organization has also been observed with acute loading of OGB-1 AM, rather than genetic expression of GCaMP (*Clancy et al., 2015*). While GCaMP expression is commonly used to study cortical plasticity, we cannot rule out that GCaMP expression affects plasticity. Only 4% of cells in Drd3-Cre mice and 1.2% of cells in Scnn1a-Cre mice showed nuclear GCaMP fluorescence indicative of GCaMP overexpression, and were excluded from analysis.

## Cell-attached recording

For simultaneous imaging and cell attached recordings, we injected a non-Cre-dependent GCaMP6s virus (AAV1-Syn-Flex-GCaMP6S-WPRE) at ~200 and ~300 µm below the dura at one stereotaxic location in S1. After allowing ~3 weeks for expression, a 2 mm diameter craniotomy was made over S1, as described for imaging experiments. Anesthesia and recording conditions were as described for imaging. We recorded juxtacellularly from GCaMP6s-expressing L2/3 neurons under 2-photon guidance using a recording pipette (3 µm tip, 3–5 MΩ) filled with fluorescent HEPES-buffered Ringers (in mM: 126 NaCl, 20 HEPES, 2.5 KCl, 2 CaCl2, 1.3 MgSO4, 14 D(+)Glucose, 50 AlexaFluor-594, pH 7.3, 290 mOsm). A loose seal configuration was obtained, and spike-associated currents were measured in voltage-clamp mode with holding potential adjusted to maintain a holding current of 0 pA. Spikes from the loose seal recording were collected simultaneous to GCaMP imaging. Whisker stimulation was as described for imaging experiments. Spikes occurring 0–100 ms after each whisker deflection were considered whisker-evoked for receptive field quantification.

## Analysis

All data analysis was conducted in MATLAB using custom-written routines unless otherwise noted. Data analysis code is available at GitHub (*LeMessurier, 2019*; copy archived at https://github.com/elifesciences-publications/imaging_analysis_pipeline) .

### Image processing and ROIs

Imaging movies were corrected for slow XY motion in Matlab using dftregistration (*Guizar-Sicairos et al., 2008*) (Matlab file exchange). We did not observe substantial Z-axis motion. Fluorescence of each pixel was smoothed in time (moving median filter, four frames). Ellipsoid regions-of-interest (ROIs) were drawn manually over neuronal somata that appeared in all movies from an imaging field. The ROI signal was the mean fluorescence of its component pixels. For neuropil subtraction, a neuropil mask was created as a 10 pixel-wide ring beginning two pixels from the somatic ROI. Neuropil pixels that were correlated with any soma ROI (with r>0.2) were removed from the neuropil mask. For L4 ROIs, mean fluorescence of the neuropil mask was scaled by 0.3 and subtracted from the raw somatic ROI fluorescence. For L2/3 ROIs, neuropil subtraction was not used in the main analysis, but is presented in *Figure 5—figure supplement 2*. Fluorescence time series of all ROIs were inspected manually to remove any movies in which mean brightness decreased >~10% due to imaging errors (e.g., loss of meniscus under the objective lens). For each ROI, fluorescence

time series were converted to ΔF/F defined as $(F_t-F_0)/F_0$, in which $F_0$ is the 20th percentile of fluorescence across the entire 80 s movie and $F_t$ is the fluorescence on each frame.

## Quantification of whisker responses, receptive fields, and response magnitude

The whisker-evoked ΔF/F signal was quantified for each ROI on each trial as (mean ΔF/F in the 1 s period following whisker train onset) – (mean ΔF/F in the 0.5 s prior to the stimulus). Each cell's 9-whisker receptive field was quantified from the median ΔF/F to each whisker across all stimulus repetitions. A ROI was considered significantly responsive to a given whisker if the distribution of evoked ΔF/F on whisker stimulus trials was significantly greater than on blank trials. This was computed using a permutation test: for each whisker, a vector of mean ΔF/F for each stimulus iteration was combined with mean ΔF/F for each blank trial (using the same frames as for measuring evoked activity). The combined distribution of means was split into two groups, and the difference in means of the two groups was measured. This was repeated 10,000 times, and the actual difference between the stimulus and blank distributions was compared to the distribution of permuted differences in means. A difference was considered significant if it was greater than the 95th percentile of the permuted distribution. P values for each of the nine whiskers were corrected for multiple comparisons by False Discovery Rate (*Benjamini and Hochberg, 1995*). A ROI was considered whisker responsive if it was significantly responsive to at least one whisker above baseline activity. For L2/3 analysis, all whisker-responsive ROIs located within any of the nine columns corresponding to the stimulated whiskers were included. For L4 analysis, only responsive ROIs located within columns C1, C2, and C3 were included to ensure similar spatial sampling of the barrel field, since imaging field locations were less overlapping between EN and NH compared to L2/3.

For comparison of the magnitude of whisker-evoked responses across imaging fields, evoked ΔF/F was normalized to ΔF/F on blank trials. This was done by computing median evoked ΔF/F across all stimulus trials, and z-scoring to the distribution of ΔF/F on blank trials. This normalization enables comparisons across imaging fields and neurons with different levels of GCaMP expression, which yield different absolute fluorescence intensity and ΔF/F signals.

## Quantification of L4 barrel size

To quantify barrel size (*Figure 2—figure supplement 2*), the area of each barrel (C1:C3, D1:D3, E1: E3) was measured from the CO-stained tangential sections. Mean radius was calculated from the area, assuming circularity. Mean barrel size for each mouse was calculated as the mean across these nine barrels.

## Normalized anatomical reference frame for spatial analysis across imaging fields

To combine imaging results across different imaging fields, ROI coordinates were transformed into a common reference frame. This was done by measuring XY position of each ROI relative to anatomical barrel centers, and rotating the coordinates for each imaging field around the column of interest to achieve the same anatomical orientation. Response or tuning measures for individual ROIs were plotted directly overlaid within this reference frame (*Figure 3E*). Alternatively, individually ROIs were spatially binned using k-means clustering so that each bin contained a similar number of ROIs, average response strength or tuning was computed for each bin, and the result was displayed as a Voronoi plot with each polygon representing a spatial bin (*Figure 5D*, *Figure 6F–G*).

## Signal and noise correlation analysis

Signal and noise correlation were computed between all pairs of simultaneously imaged, whisker-responsive neurons that were located within (for L4 cells) or above (for L2/3 cells) any L4 barrel. Cells located within or above L4 septa were not used for correlation analysis, to avoid ambiguity in localizing the precise barrel boundary. Thus, closely spaced cross-column cell pairs were located in adjacent barrel columns, not in column and adjacent septum. To compute signal correlation, we calculated for each ROI a 9-element vector composed of its mean response to each whisker over all stimulus repetitions. Each vector was individually Z-scored. Signal correlation was defined as the Pearson's correlation of the vectors for a pair of neurons. To compute noise correlations, we

constructed for each ROI, nine vectors (one for each whisker) containing the mean ΔF/F during the evoked window following each stimulus iteration. Each vector was individually z-scored. We computed the Pearson's correlation for each pair of neurons separately for each whisker, then computed the mean correlation for all whiskers. Distance between neurons was calculated as the Euclidean distance (μm).

### Neural decoder

We constructed neural decoders to detect a whisker deflection compared to spontaneous activity from single-trial mean ΔF/F of individual ROIs and ensembles of ROIs. Each ROI was represented by a nine binary classifiers – one per whisker – that were trained by logistic regression to report the probability of a stimulus given the mean ΔF/F during the evoked window following a single whisker deflection, selected randomly from each stimulus iteration (structured as in *McGuire et al., 2016*). For each logistic function coefficients were fit by logistic regression and K-fold cross validation to relate the mean ΔF/F on each trial to the probability of given whisker having been deflected. Each iteration of model fitting was performed on 80% of trials (including the given whisker and blank trials), and performance was assessed on the remaining 20% of trials. For single ROI decoders, predictions were made by selecting the highest probability on a given trial. The fitting and testing process was repeated 500 times for each single ROI decoder, and performance was averaged across iterations.

For ensemble decoding, simultaneously-imaged ROIs were clustered into ensembles either randomly or based on position within the imaging field using k-means clustering. To sample a range of ensemble sizes and positions, clustering was performed for each field while varying the number of ensembles between two to as many ensembles as responsive ROIs. This was repeated separately for each whisker, and positions used for clustering were normalized relative to the reference whisker. This yielded ensemble sizes of 2:42, with 4242 ± 1400 total ensembles per NH field, and 9630 ± 1501 ensembles per EN field. For ensemble decoders composed of random sets of ROIs, we trained and tested ensemble decoders of 1, 2, 3, 5, 10, 15, 20, 25, and 30 ROIs located within the same imaging field. For each ensemble size, we tested the lesser of 500 or N-choose-K ensembles (where N is the total number of responsive ROIs in the field and K is the ensemble size) randomly drawn from each imaging field. We used a maximum of 30 ROIs per field, in order to include as many imaging fields as possible in this analysis. To predict single-trial stimuli from ensembles, the output of each ROI classifier was normalized so that each unit had the same weight in population decoding. The population stimulus prediction was calculated by summing the probabilities of each trial type (stimulus or blank) over all units in an ensemble and selecting the trial type with the maximal summed probability.

## Acknowledgements

We acknowledge the GENIE project (HHMI Janelia Research Campus) for GCaMP6s virus, and thank Paley Han for mouse colony management and Katherine Smith for experimental support. This work was supported by NIH R37 NS092367.

## Additional information

### Funding

| Funder | Grant reference number | Author |
|---|---|---|
| National Institute of Neurological Disorders and Stroke | R37NS092367 | Daniel E Feldman |

The funders had no role in study design, data collection and interpretation, or the decision to submit the work for publication.

### Author contributions

Amy M LeMessurier, Conceptualization, Software, Formal analysis, Investigation, Methodology, Writing—original draft, Writing—review and editing, Designed and conducted experiments, Wrote

MATLAB analysis software, Analyzed data, Co-wrote manuscript; Keven J Laboy-Juárez, Software, Methodology, Contributed to design of neural decoder, Provided software; Kathryn McClain, Formal analysis, Methodology, Developed methods for calcium imaging data analysis; Shilin Chen, Investigation, Methodology, Performed histology, Contributed to anatomical reconstruction of cell locations; Theresa Nguyen, Investigation, Methodology, Performed histology, Developed methods for reconstruction of cell locations; Daniel E Feldman, Conceptualization, Supervision, Funding acquisition, Writing—review and editing, Planned and designed experiments, Wrote Igor data acquisition software, Co-wrote manuscript

#### Author ORCIDs
Amy M LeMessurier (ID) https://orcid.org/0000-0002-5119-0524
Daniel E Feldman (ID) https://orcid.org/0000-0003-4646-8170

#### Ethics
Animal experimentation: All animal procedures were reviewed and approved by UC Berkeley's Institutional Animal Care and Use Committee, protocol AUP-2016-02-8351-1.

#### Decision letter and Author response
Decision letter https://doi.org/10.7554/eLife.46321.023
Author response https://doi.org/10.7554/eLife.46321.024

## Additional files

#### Supplementary files
• Transparent reporting form
DOI: https://doi.org/10.7554/eLife.46321.018

#### Data availability
Data are available on CRCNS data sharing website, www.crcns.org, under the DOI http://doi.org/10.6080/K0DB801M. Users must first create a free account (https://crcns.org/register) before they can download the datasets from the site. Data analysis code has been made available on GitHub at (https://github.com/alemessurier/imaging_analysis_pipeline; copy archived at https://github.com/elifesciences-publications/imaging_analysis_pipeline).

The following dataset was generated:

| Author(s) | Year | Dataset title | Dataset URL | Database and Identifier |
| --- | --- | --- | --- | --- |
| Amy M LeMessurier, Daniel E Feldman | 2019 | 2-photon calcium imaging in S1 of enriched and normally housed mice during individual deflection of 9 whiskers | http://doi.org/10.6080/K0DB801M | CRCNS - Collaborative Research in Computational Neuroscience, 10.6080/K0DB801M |

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
