## [Decision Letter]

Thank you for submitting your article "Tactile enrichment drives emergence of functional columns and improves sensory coding in the whisker map in mouse S1" for consideration by *eLife*. Your article has been reviewed by two peer reviewers, and the evaluation has been overseen by a Reviewing Editor and Laura Colgin as the Senior Editor. The following individual involved in review of your submission has agreed to reveal their identity: Michael Brecht (Reviewer #1).

The reviewers have discussed the reviews with one another and the Reviewing Editor has drafted this decision to help you prepare a revised submission.

The reviewers and editors were impressed with the scope and detail of the work, and consider this an important study addressing the impact of an animal's experience on sensory maps in the cerebral cortex.

The reviewers agreed on the following key requests to be addressed in the revision:

1) Clarify the enrichment conditions (here, please make use of the generous space at *eLife* by including additional figures, or videos, and descriptions of the enrichment setting), and discuss beyond-tactile effects.

2) Address the methodological concerns / confounds raised by both reviewers, in particular the genetic strain and effects of the health of the animal on exploratory behavior. This would ideally be supported by additional analyses / control data.

Please consider all additional comments by the reviewers as suggestions for improving the manuscript. The individual reviews are included in their entirety below.

*Reviewer #1:*

In their paper 'Tactile enrichment drives emergence of functional columns and improves sensory coding in the whisker map in L2/3 of mouse S1' LeMessurier et al. describe effects of enrichment whisker tuning in the mouse barrel cortex. The authors’ house mice either in enriched or normal home cages and assess response properties with 2-photon Ca^2+^ imaging. The authors find more sharply delineated functional columns and a higher degree of whisker selectivity as a result of enrichment. My views on the paper are the following:

1) This is a good study. Enrichment effects have been studied before in barrel cortex and other cortices, but the authors go well beyond, what has been done previously. The current study achieves this novel depth of analysis by 2-photon imaging and careful analysis of the data.

2) The authors’ should describe their enrichment procedures in more detail. In the current version of the manuscript these procedures mainly figure in just one subpanel. Having said that more detail on enrichment procedures should be provided, I would also like to add that I am not sure that the effects described here are a result of 'tactile enrichment'. If I got it right multiple parameters differed between enriched and non-enriched conditions (cagemates’ toys etc.). Since it is not clear which parameter drove the changes described here, it might be indicated to drop the term 'tactile' from the title.

3) It would be great if the authors could comment on the effects of enrichment on directional tuning. Such an analysis would be most welcome, because the organization of directional tuning in barrel cortex is controversial. I am not sure, however, if the authors’ stimulus design will allow such an analysis post hoc. If such an analysis is therefore not possible, I would not require a redo of the experiments to answer this question.

4) The authors might want to discuss more broadly and prominently the specifics of the transgenic mouse lines and the potential side effects of GCamp expression and transgenic procedures on their results. Thus, rather than merely referring to neurons as L4-cells it might be necessary to specify the Cre-line used for targeting.

*Reviewer #2:*

This paper directly addresses a lurking, unproven hypothesis, that more diverse, natural and/or behaviorally-relevant vibrissal use increases the systematicity/smoothness of vibrissa map organization in L2/3. Whilst data from Polley et al., 2004 using an extreme enrichment environment somewhat implies this result, their methods did not allow for laminar precision, and the extremity of the conditions used made the findings inconsistent with implementation in typical barrier facilities. Also, that finding was in rats, a far less relevant model given current bias' in model animal selection for vibrissal studies, and some of the findings here directly disagree with conclusions in that study.

These present results are particularly important because this model is widely used to understand the connection between plasticity and brain organization. This finding impacts the interpretation of hundreds of prior papers, and will substantially change procedures used in subsequent studies (or, future authors will be required to indicate why they have not made such a change). The present results are also really nicely collected, analyzed and presented – put simply, they make an important finding easy to understand.

1) Did enriched mice survive the initial viral injection surgery more robustly than the non-enriched (probability of death or complications, rate of weight re-gain post-surgery, amount of time before being returned to the home cage and fellow mouse partners, anything else?) The amount of whisker use will directly relate to the health of the animal post-surgery, posing a potential confound unrelated to enrichment per se (except insofar as healthy mice are, well, healthy). I can imagine many kinds of data analysis and straightforward double-blinded surgery experiments that would satisfy this concern, but something specific should be done to address it.

2) Were the anatomical L4 barrels themselves different in any way? Diameter/size, smoothness of the wall, number or size of sub-zones (ala Land's work), anything else? The effects are interesting either way, but it is a major source of information in the existing data that could be robustly related (or, shown unimportant) to the functional changes.

3) Were the mice left solo after their cage mate was taken (and sacrificed) also used? If so, is there a relationship between map patency and how long after the removal of imaged mouse #1 that imaged mouse #2 was imaged? This might give at least a clue as to the import of social touch in the effects observed.

4) Z-scoring to baseline begs the question of whether there were differences in baseline activity levels (and, therefore, 'noise' in the estimate). Were there differences?

---

## [Author Response]

The reviewers agreed on the following key requests to be addressed in the revision:1) Clarify the enrichment conditions (here, please make use of the generous space at eLife by including additional figures, or videos, and descriptions of the enrichment setting), and discuss beyond-tactile effects.

We now describe enrichment conditions in detail (Results and Materials and methods), including in new Figure 1—figure supplement 1. Because beyond-tactile effects may contribute to our effects, we have changed the title to say that “enrichment” drives the emergence of columnar structure, rather than “tactile enrichment”. The Introduction and Discussion were similarly updated. See reviewer 1 point 2 for details.

2) Address the methodological concerns / confounds raised by both reviewers, in particular the genetic strain and effects of the health of the animal on exploratory behavior. This would ideally be supported by additional analyses / control data.

The first concern was whether GCaMP expression or transgenic procedures influenced the results (reviewer 1). We addressed this in several ways – by showing that GCaMP overexpression (i.e. nuclear expression) was very low; by pointing out that basic salt-and-pepper map topography has been observed without GCaMP by using non-genetic expression of OGB-1 AM; and by clearly presenting the mouse genetic backgrounds. While there are some differences in genetic background, both Drd3-Cre and Scnn1a-Cre lines are on primarily a C57BL/6 background, so we think that comparison of map topography between these lines is reasonable. See reviewer 1 point 4.

The second concern was whether there were differences in post-surgical health between enriched and normally housed mice, which could have led to altered exploration and thus different extents of use-dependent map sharpening (reviewer 2). We addressed this by quantifying surgical recovery in a new cohort of EN and NH mice. We now show that surgical recovery time was similar between EN and NH mice (4-5 days), and was brief compared to the total EN or NH exposure time (35-40 days). We also show that spontaneous behavior was similar between EN and NH groups during this surgical recovery period. This suggests that differences in surgical recovery are not responsible for the differences in map topography. See reviewer 2 point 1.

Please consider all additional comments by the reviewers as suggestions for improving the manuscript.

We have addressed many of these, as spelled out below.

The individual reviews are included in their entirety below.Reviewer #1:In their paper 'Tactile enrichment drives emergence of functional columns and improves sensory coding in the whisker map in L2/3 of mouse S1' LeMessurier et al. describe effects of enrichment whisker tuning in the mouse barrel cortex. The authors’ house mice either in enriched or normal home cages and assess response properties with 2-photon Ca^2+^ imaging. The authors find more sharply delineated functional columns and a higher degree of whisker selectivity as a result of enrichment. My views on the paper are the following:1) This is a good study. Enrichment effects have been studied before in barrel cortex and other cortices, but the authors go well beyond, what has been done previously. The current study achieves this novel depth of analysis by 2-photon-imaging and careful analysis of the data.

Thank you.

2) The authors’ should describe their enrichment procedures in more detail. In the current version of the manuscript these procedures mainly figure in just one subpanel. Having said that more detail on enrichment procedures should be provided, I would also like to add that I am not sure that the effects described here are a result of 'tactile enrichment'. If I got it right multiple parameters differed between enriched and non-enriched conditions (cagemates’ toys etc.). Since it is not clear which parameter drove the changes described here, it might be indicated to drop the term 'tactile' from the title.

We now describe enrichment in detail, both in the Results (subsection “Enrichment sharpens the point representation of a single whisker in L2/3”) and Materials an methods (subsection “Enrichment”), and we have added new Figure 1—figure supplement 1 that shows the enrichment toys and how they were combined in enriched cages, and shows example enriched and normally-housed cages.

We agree that enrichment constitutes more than just increased tactile experience, and that changes in the whisker map may be driven by a combination of sensory experience, increased social interactions, greater activity levels, and other factors. We now discuss this explicitly in Results: “This enriched environment will enhance not only tactile experience, but also social interaction, activity level, and other factors, all of which could drive cortical plasticity.” And in Discussion: “While enrichment promotes more diverse tactile experience, it also likely increases social interactions and affects general activity levels. Which of these factors is dominant in driving whisker map plasticity was not tested.”

To reflect the multi-faceted nature of enrichment, we have changed the title from effects of “tactile enrichment” to simply “enrichment”. This does not weaken the overall conclusions of the paper, which remain that salt-and-pepper, dispersed sensory maps partially reflect inadequate experience in rodent laboratory housing.

3) It would be great if the authors could comment on the effects of enrichment on directional tuning. Such an analysis would be most welcome, because the organization of directional tuning in barrel cortex is controversial. I am not sure, however, if the authors’ stimulus design will allow such an analysis post hoc. If such an analysis is therefore not possible, I would not require a redo of the experiments to answer this question.

This is a great suggestion, but unfortunately we did not apply different directions of whisker stimuli, so we cannot assess direction tuning in our data set. Prior papers on the direction map indicate that age and/or experience is critical to create a discernible direction map in L2/3, and a future study that clarifies this at single-cell resolution would be nice.

4) The authors might want to discuss more broadly and prominently the specifics of the transgenic mouse lines and the potential side effects of GCamp expression and transgenic procedures on their results. Thus, rather than merely referring to neurons as L4-cells it might be necessary to specify the Cre-line used for targeting.

This is an important point which we had not emphasized enough in the first version. We now describe the transgenic mouse lines more completely, and discuss possible impacts of the transgenes and background strain differences on map organization. To emphasize to readers that L2/3 and L4 cells were defined by the Drd3-Cre and Scnn1a-Cre transgenes, rather than just by layers, we re-state at the beginning of each major Results section that L2/3 cells were from Drd3-Cre mice and L4 cells were from Scnna1-Cre mice.

Possible effects of background strain. Drd3-Cre mice were obtained from MMRRC on a mixed C57BL6/FVB background, and were crossed to C57BL/6 mice for several generations before generating the mice used in these experiments. Scnn1a-Cre mice were obtained on a C57BL/6 background from JAX and maintained on a C57BL/6 background. Thus the genetic backgrounds of both strains are similar (primarily C57BL/6). Drd3-Cre mice were slightly larger than Scnn1a-Cre mice, and had somewhat larger anatomical barrels (now shown in Figure 2—figure supplement 2), indicating that some strain differences do exist. However, these differences in anatomical barrel size (by ~25 μm) were much smaller than the differences in point representation between L2/3 and L4 (Figure 2, Figure 3, Figure 3—figure supplement 1). To show the reader these differences (and that they were small in scale), we explicitly plot the mean anatomical barrel size overlaid on the functional maps in the key figures (Figure 2 all panels, Figure 5E-F).

Possible effects of transgenes. The Drd3-Cre transgene does not appear to strongly impact basic somatotopic map organization in L2/3, because we previously observed salt-and-pepper organization in L2/3 in wild-type C57BL/6 mice (Clancy et al., 2015) that was essentially identical to that observed here in Drd3-Cre mice. We do not know if the Scnn1a-Cre transgene perturbs L4 map organization. We now discuss these issues explicitly in Materials and methods (subsection “Animals”). The Discussion also now states the key premise in this study design: “This comparison between L4 and L2/3 assumes that Drd3-Cre and Scnn1a-Cre mice have overall similar somatotopic map organization.”

Possible side effects of GCaMP expression. We previously observed salt-and-pepper organization in L2/3 using acute loading of OGB-1 AM, rather than GCaMP (Clancy et al., 2015). Thus, basic map organization is not an artifact of GCaMP expression. While it is possible that GCaMP expression impacts plasticity, GCaMP expression is commonly used in cortical plasticity studies. Only 4% of cells in Drd3-Cre mice and 1.2% of cells in Scnn1a-Cre mice showed nuclear GCaMP fluorescence indicative of GCaMP overexpression, and were excluded from analysis. Thus, GCaMP overexpression was not a major problem in this study. We now discuss these issues in Materials and methods (subsection “2-photon calcium imaging”).

Overall, these issues are now discussed much more clearly, and while some strain differences may be present, we don’t think these account for the functional differences in whisker maps between layers.

Reviewer #2:1) Did enriched mice survive the initial viral injection surgery more robustly than the non-enriched (probability of death or complications, rate of weight re-gain post-surgery, amount of time before being returned to the home cage and fellow mouse partners, anything else?) The amount of whisker use will directly relate to the health of the animal post-surgery, posing a potential confound unrelated to enrichment per se (except insofar as healthy mice are, well, healthy). I can imagine many kinds of data analysis and straightforward double-blinded surgery experiments that would satisfy this concern, but something specific should be done to address it.

Surgical morbidity and mortality were low and equivalent between NH and EN groups. Including all mice (even those with cloudy cranial windows or low GCaMP expression, where imaging could not be performed), 5.5% of NH mice and 4.8% of EN mice had post-surgical mortality. No mice had morbidity sufficient to remove them from the study. Immediate post-surgical recovery from anesthesia was fast, with mice returned to the home cage within ~ 1 hr post-surgery, once they recovered posture and locomotion. This immediate recovery appeared similar between NH and EN groups.

Slower phases of surgical recovery were quantified in a new cohort of 6 NH mice and 6 EN mice. These mice received standard headplate and cranial window surgery, but not viral injection. NH and EN mice recovered to pre-surgical weight within 4 and 5.5 days, respectively (Author response image 1). The slight delay for EN mice may reflect difficulty learning to navigate the enriched environment while carrying imaging headgear. We examined general exploratory behavior during this recovery period by placing each mouse, daily, in a clean cage with bedding but no toys or nesting material. After a 10-min habituation period, we quantified time spent in 3 spontaneous behaviors (locomoting, grooming, and rearing) for a 10 min observation period. The prevalence of these behaviors did not differ between EN and NH mice, although EN mice showed a non-significant trend for more locomotor behavior (Author response image 1). Behavior in the home cage could not be quantified because enrichment objects and the cage top obscured observation. Thus, these measurements suggest that exploratory behavior during the recovery period was roughly similar between EN and NH mice.

**Author response image 1. respfig1:** Comparison of surgical recovery between NH and EN mice. Data are from a new cohort of 6 NH and 6 EN mice, all C57BL/6 wild-types. Surgery for headplate and cranial window implant was on Day 0. A. Daily weight during surgical recovery. Mean recovery to pre-surgical weight was 4 days (NH) or 5.5 days (EN). B. Spontaneous behaviors quantified in a clean open-field environment. Time spent in each behavior was calculated with 30-sec resolution. ‘Still’ indicates 30-sec periods in which neither locomotion, grooming or rearing occurred. Behavior was recorded daily and analyzed in 2-day windows. Circles and bars are bootstrapped medians and 95% confidence intervals. P-values are for permutation test for difference between NH and EN in total area under the curve. There were no significant differences between NH and EN in any behavior (α = 0.05).

We now report these findings in Materials and methods (subsection “Surgery”). We explicitly report that the duration of surgical recovery (4-5.5 days) was brief relative to the total duration of NH or EN before imaging (35-40 days, Table 1). Overall, we believe that surgical recovery was short enough, and similar enough between NH and EN groups, that whisker map changes are more likely attributable to the NH or EN environment, rather to differences in surgical recovery per se in that in environment.

2) Were the anatomical L4 barrels themselves different in any way? Diameter/size, smoothness of the wall, number or size of sub-zones (ala Land's work), anything else? The effects are interesting either way, but it is a major source of information in the existing data that could be robustly related (or, shown unimportant) to the functional changes.

This is a good question, especially because Polley et al., 2004 reported subtle enlargement of L4 barrels using a vigorous enrichment paradigm in adult rats. To test if a similar effect occurred in our mice, we analyzed flattened tangential CO-stained sections from all mice in which all 9 barrel columns (C1:C3, D1:D3, E1:E3) had been sampled. We calculated the radius of each barrel. Enrichment did not alter mean barrel radius in either Drd3-Cre mice or Scnn1a-Cre mice. This is now reported in Results (subsection “Enrichment sharpens the point representation of a single whisker in L2/3”, last paragraph), and new Figure 2—figure supplement 2.

We did not analyze wall smoothness or sub-barrel patterns (Land), because in our hands they could not be reliably enough visualized or quantified. Please note that Land and Erickson (J. Comp. Neurol. 2005) report that mice lack sub-barrel structures, unlike rats. Thus, the functional changes that we observed in the whisker map occurred without gross changes in barrel anatomy, although we cannot rule out subtle structural plasticity not observable in average CO staining patterns.

3) Were the mice left solo after their cage mate was taken (and sacrificed) also used? If so, is there a relationship between map patency and how long after the removal of imaged mouse #1 that imaged mouse #2 was imaged? This might give at least a clue as to the import of social touch in the effects observed.

We did not remove cage mates, because we wanted to maintain a constant social environment over the duration of the experiment. Instead, when one co-housed mouse was finished with imaging, it remained in the shared cage until all mice in that cage were finished with imaging. Then all mice were sacrificed and processed for histology. This is now stated in Results (subsection “2-photon calcium imaging”). We agree it would be interesting to examine the impact of social touch in particular on enrichment effects, but we don’t have the data to do this here.

4) Z-scoring to baseline begs the question of whether there were differences in baseline activity levels (and, therefore, 'noise' in the estimate). Were there differences?

We reported in Figure 4C that spontaneous activity (measured as median ΔF/F on blank trials) was decreased in L2/3 of EN mice relative to NH mice. In response to this reviewer comment, we tested whether the reduction in spontaneous activity could fully explain the increase in CW-evoked response magnitude or sharpening of whisker receptive fields, both of which were quantified in Figure 4A-B as whisker-evoked ΔF/F Z-scored to spontaneous activity. In the new Figure 4—figure supplement 1, we analyze response magnitude and receptive field sharpness using absolute ΔF/F, not Z-scored to spontaneous. By comparing mean absolute ΔF/F to the 4 strongest and 4 weakest whiskers for each neuron, we found that the mean response to the 4 strongest whiskers was greater in EN than NH mice, and the slope relating strongest- to weakest-whisker responses was greater in EN mice (Figure 4—figure supplement 1). Thus, EN increases CW-response magnitude and sharpens receptive fields without Z-scoring to spontaneous.

Z-scoring to spontaneous provides a convenient measure of ‘signal-to-noise’ for sensory responses, and helps normalize for varying GCaMP expression levels across cells. Our presentation of Z-scored measures of responsiveness and tuning in Figure 4A-B is designed to estimate signal-to-noise in sensory coding. The neural decoders (Figure 7) test explicitly whether signal-to-noise is improved in EN mice, by predicting presence or absence of a whisker deflection from single-trial absolute (non-Z-scored) ΔF/F from either whisker stimulus trials or blank trials. The decoder analysis showed enhanced ability to detect CW deflection vs. blank from EN mice (Figure 7C), with a spatial profile (Figure 7D) that is consistent with the mean tuning changes reported in Figure 4B. Thus, our results are internally consistent, and show that EN both modestly increases absolute CW-evoked ΔF/F and reduces spontaneous activity, and that together these increase signal-to-noise of the sensory code to improve stimulus detection.